# Topology- and Distribution-aware Backdoor Defense Against Federated Graph Learning

## Abstract

Federated graph learning (FGL) has rapidly gained prominence as a privacy-preserving collaborative paradigm. However, the increasing prevalence of backdoor attacks presents significant challenges to federated systems. These attacks rely on the injection of carefully crafted triggers that lead to erroneous predictions. Recent research has shown that the diversity of trigger structures and injection locations in FGL diminishes the effectiveness of traditional federated defense methods. Notably, existing defense strategies for FGL have yet to fully exploit the unique topological structures of graphs, highlighting opportunities for improvement in countering these attacks.

To this end, we propose a tailored topology- and distribution-aware backdoor defense against federated graph learning method (FedTD). At the client level, we introduce an energy function to integrate the underlying data distribution into the local model, assigning low energy to benign clients and high energy to malicious clients. By combining topological features with the energy function, we establish a more comprehensive energy estimation. At the server level, we construct a virtual graph based on estimation of each client to evaluate the maliciousness score of each client. The homophily level of each local graph is considered to ensure the reliability of the virtual graph. During aggregation, we assign lower weights to clients with high malicious scores and higher weights to clients with low malicious scores, thus achieving a more robust FGL. FedTD remains robust under both small and large malicious client ratios. Extensive results across various federated graph scenarios under backdoor attacks validate the effectiveness of FedTD. The code is anonymous available[1].

## 1 Introduction

Federated Learning (FL) (Yang et al., 2019; McMahan et al., 2017) enables distributed collaborative machine learning by allowing multiple clients to jointly train a shared global model while preserving the privacy of sensitive data. This eliminates the need for aggregating distributed data and ensures compliance with privacy protocols (Wei et al., 2020a). Recently, some studies have focused on training Graph Neural Networks (GNNs) (Kipf, 2016; Wu et al., 2020) on isolated graph data using FL, a paradigm referred to as Federated Graph Learning (FGL) (Liu et al., 2024a). FGL presents a promising solution for distributed graph-structured scenarios, such as epidemiology (Liu et al., 2024b) and scene graphs (Wei et al., 2020b; Xu et al., 2025b).

While the distributed paradigm of FL offers many benefits, it also introduces additional vulnerabilities, particularly in the form of backdoor attacks from malicious participants (Li et al., 2022; Bagdasaryan et al., 2020). These attacks involve injecting harmful data or models during training, embedding hidden behaviors that are activated under specific conditions to trigger erroneous model outputs. The goal of such attacks is to cause local models to learn incorrect information and activate the backdoor at critical moments, leading to inaccurate predictions. In traditional FL setting, extensive studies have provided mature defense method against backdoor attacks (Gong et al., 2022; Lyu et al., 2022). Specifically, some studies (Shejwalkar & Houmansadr, 2021; Ozdayi et al., 2021) exclude outlier updates based on the statistical properties of model outputs, while others (Pillutla et al., 2022) identify

---

[1] https://anonymous.4open.science/r/FedTD-r

malicious clients by measuring statistical differences between local models or between local and global models.

However, the complex graph structures and highly diverse feature information in FGL setting limit the effectiveness of existing backdoor defense methods. A few recent study (Huang et al., 2024) has explored backdoor defenses in graph classification tasks, but defenses for backdoor attacks in node classification tasks remain underdeveloped. More recently, FedTGE (Wan et al., 2025) pioneered exploration into backdoor defenses for node classification tasks. It uses an energy function combined with graph convolution to model clients' data distributions, thereby identifying malicious clients. However, two significant limitations still persist: **1)** *The difference in graph homophily levels among different clients influences with the modeling of data distributions and the identification of triggers.* **2)** *Graph topological features is not sufficiently considered.* We further introduce on these two limitations in detail.

**1)** *The difference in graph homophily levels among different clients influences with the modeling of data distributions and the identification of triggers.* Recent study (Tan et al., 2025) has revealed that the homophily level of clients in the FGL setting significantly impacts the differences among clients. Specifically, the homophily level of a client is defined as the proportion of homophily edges within its graph. An homophily edge is defined as an edge connecting two nodes with the same label, while a heterophily edge connects two nodes with different labels. Based on this definition, the set of all edges $\mathcal{E}$ in a graph $\mathcal{G}$ can be divided into two subsets: the homophily edge set $\mathcal{E}_{ho}$ and the heterophily edge set $\mathcal{E}_{he}$. The homophily level is then defined as the proportion of homophily edges in the graph: $\frac{\mathcal{E}_{ho}}{\mathcal{E}_{ho}+\mathcal{E}_{he}}$. We provide an intuitive illustration of this concept in Figure 1 (a). Unlike the traditional FL setting, the homophily level of clients in the FGL setting affects the distributional differences among clients, as noted in previous study (Tan et al., 2025). In backdoor attacks, the difference between homophily and heterophily edges increase the difficulty for defense methods to identify triggers. Triggers with the same shape and size can exhibit diverse distributional differences due to the differing nature of edges (refer to Figure 1 (b)). Therefore, incorporating the homophily level into the design of backdoor defense methods is crucial.

**2)** *Graph topological features is not sufficiently considered.* Previous study (Wan et al., 2025) has simply utilized GCNs to aggregate energy within graphs, thereby modeling the data distribution of each client's entire graph. However, the topological features of client graphs (e.g., node degree, clustering coefficient, centrality, etc.) have not been sufficiently explored (refer to Figure 1 (c)), resulting in a lack of consideration for graph topology. Moreover, directly using GCNs to aggregate energy within

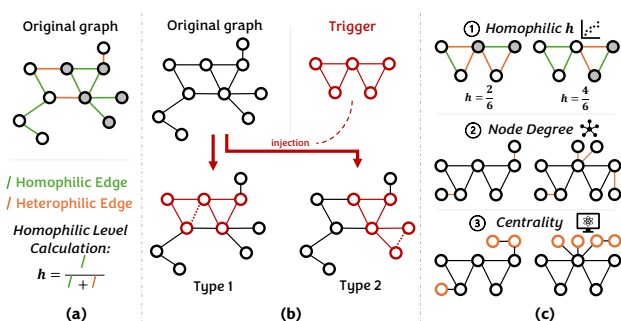

Figure 1: Problem illustration.

graphs fails to distinguish between homophily and heterophily edges. For homophily edges, the connected nodes are expected to have similar features, while for heterophily edges, the connected nodes exhibit feature dissimilarity. These topological features are highly correlated with the shape and size of triggers. Therefore, incorporating topology into the design of backdoor defense methods is crucial for enhancing their robustness and effectiveness.

To address these two significant limitations, we propose a tailored topology- and distribution-aware backdoor defense against federated graph learning method (FedTD). Specifically, we introduce an tailored Topology- and Distribution-aware Client Estimation component (TDCE). TDCE includes an energy function with GCN that incorporates the underlying data distribution into the local model. This function assigns low energy to benign samples and high energy to constructed malicious substitute samples. By combining topological features with the energy function, we establish a more comprehensive energy estimation for each client. Furthermore, we introduce a tailored Topology- and Distribution-aware Graph Construction component (TDGC). TDGC calculates the estimation differences between each pair of clients and constructs edges between nodes with high similarity in their estimations, thereby forming a new virtual graph. Notably, beyond simply utilizing distributional

information and topological features for similarity calculation, TDGC additionally incorporates the homophily level to achieve more fine-grained identification of malicious clients. Furthermore, TDGC leverages the structural features of the virtual graph to evaluate the maliciousness score of each client. Nodes with low degrees indicate their deviation from the majority of clients, implying a higher maliciousness score. During the aggregation process, clients with higher maliciousness scores are assigned lower weights, while clients with lower maliciousness scores are assigned higher weights, thereby enabling more robust FGL. Through these two carefully designed components, our FedTD effectively addresses the aforementioned two limitations.

- We identify significant limitations of existing backdoor attack defense methods in FGL setting for node classification task.

- We propose FedTD, an novel method that effectively defense backdoor attacks in FGL setting via both topological and distribution features. Our FedTD enables clients to model the data distribution and topological features of each client, considering both malicious scores and homophily level during clients aggregation.

- We conducted comprehensive experiments on five widely-used datasets under both IID and Non-IID settings, with varying proportions of malicious clients. The results show that our FedTD significantly outperforms the current state-of-the-art baselines.

## 2 RELATED WORK

### 2.1 FEDERATED GRAPH LEARNING

Federated Graph Learning (FGL) (Wan et al., 2025; Fu et al., 2022; Yue et al., 2024; Li et al., 2024b; Cai et al., 2024; Fu et al., 2025) is a decentralized graph learning paradigm that combines the characteristics of Federated Learning (FL) (Yang et al., 2019; McMahan et al., 2017; Zhao et al., 2018; Li et al., 2021a) and Graph Neural Networks (GNNs) (Xu et al., 2024; 2025a; Han et al., 2022; Shi & Rajkumar, 2020; Yang et al., 2021; Wang et al., 2024). It enables collaborative learning across multiple clients with graph-structured data while preserving data privacy. In recent years, a significant amount of studies on FGL have focused on improving the performance of global and local models (Li et al., 2024b; Cai et al., 2024; Fu et al., 2025), yet they have largely overlooked the risks of backdoor attacks introduced by graph structures and decentralization. Although extensive efforts have been devoted to defending against backdoor attacks in FL (Yazdinejad et al., 2024; Hu et al., 2024; Chen et al., 2024; Hallaji et al., 2024; Yazdinejad et al., 2024), the complex topological features of graph data and the additional client heterogeneity in the FGL setting make these methods less effective. Similarly, existing backdoor defense methods (Zhang et al., 2021b; 2024; Alrahis et al., 2023; Dong et al., 2025) for GNNs also show suboptimal performance under the FGL setting. In recent years, only FedTGE (Wan et al., 2025) has explored backdoor defenses specifically for FGL. However, FedTGE primarily focuses on data distribution while failing to fully leverage the topological features of graphs. Our work is the first to propose a robust FGL method that effectively defends against backdoor attacks by simultaneously utilizing topological features and data distribution.

### 2.2 GRAPH ROBUST LEARNING

Some traditional graph robust learning methods provide verifiable guarantees for node and graph classification predictions. For example, (Zugner & Gunnemann, 2019; Wang et al., 2021) use graph structural information to bound adversarial perturbations and propose certifiable GNN training. DropEdge (Rong et al., 2020) improves robustness and generalization by randomly dropping edges for structural smoothing. RES (Lin et al., 2023) introduces certifiable robustness for graph contrastive learning under structural perturbations. DiffSmooth (Zhang et al., 2023) enhances robustness by smoothing representations through diffusion/denoising processes. However, these methods highly rely on centralized graph learning and are difficult to be used in federated graph learning scenarios.

### 2.3 BACKDOOR DEFENSE IN FEDERATED LEARNING

Backdoor attacks pose a significant threat to FL due to their decentralized paradigm (Kapoor & Kumar, 2024; Uddin et al., 2025). To mitigate this threat, various defense methods have been

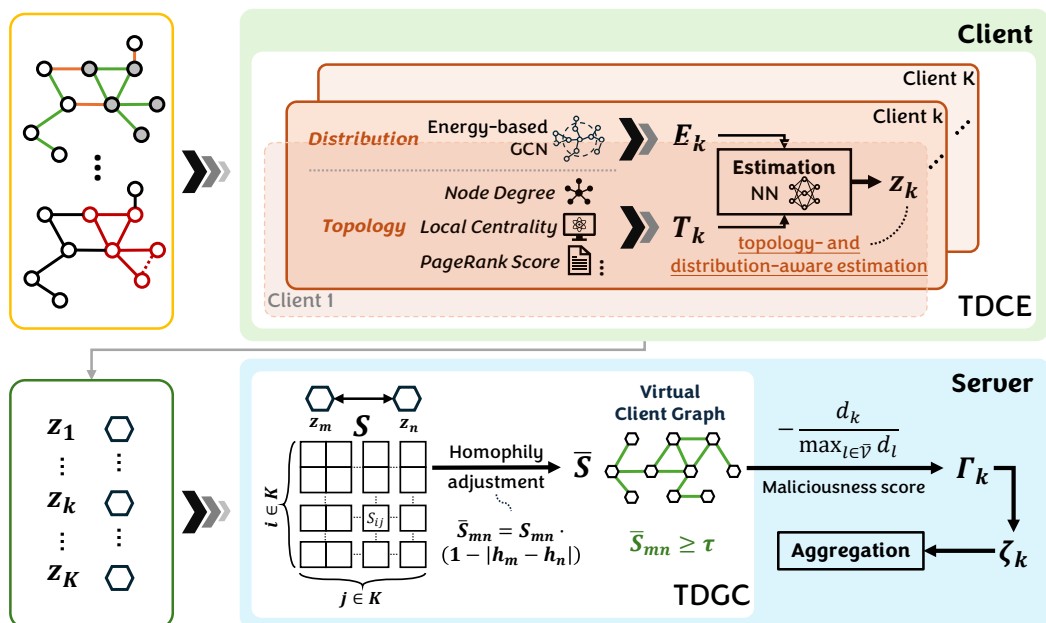

Figure 2: The overview of FedTD, which consists of two main components: TDCE and TDGC.

proposed, which can be broadly categorized into backdoor detection and backdoor removal methods. Backdoor detection aims to identify whether a model or input has been compromised. Techniques such as analyzing input deviations in the feature space (Tran et al., 2018; Chen et al., 2018; Liu et al., 2022) or detecting prediction anomalies on test inputs (Gao et al., 2019) have been widely explored. Reverse engineering methods, such as neural cleanse (Wang et al., 2019), reconstruct the trigger and identify the target label of the attack. On the other hand, backdoor removal methods focus on eliminating backdoors from compromised models while preserving their performance on benign inputs. Methods like fine-pruning (Liu et al., 2018) and neural attention distillation (Li et al., 2021c) fine-tune the model using clean inputs, while methods such as adversarial unlearning of backdoors via implicit hypergradient (Zeng et al., 2021) and anti-backdoor learning (Li et al., 2021b) adjust training processes to ensure robustness. In the FL setting, additional challenges arise due to client heterogeneity and data decentralization. Defense methods like vector filtering (Shejwalkar & Houmansadr, 2021; Guerraoui et al., 2018; Pillutla et al., 2022) effectively mitigate malicious client contributions by aggregating updates robustly. Furthermore, some studies (Park et al., 2021; Cao et al., 2020) leverage proxy data and utilize server-side knowledge to enhance robustness.

However, compared to traditional FL, FGL is more vulnerable to backdoor attacks due to the more complex topological features of graph data and the increased heterogeneity among clients, which result in more diverse and harder-to-detect triggers. Although a few studies (Wan et al., 2025; Huang et al., 2024; Yang et al., 2024) have explored backdoor defenses in FGL, they primarily rely on data distribution to identify malicious clients, lacking sufficient utilization of topological features. Our work is the first to integrate both topological features and data distribution, achieving a robust FGL.

## 3 METHODOLOGY

The overall framework of FedTD is depicted in Figure 2, with the corresponding algorithmic pseudocode provided in Algorithm 1 in Appendix A.2. In the following subsections, we first introduce the preliminaries and then detail all key components.

### 3.1 PRELIMINARY

**Definition.** We follow the general paradigm of federated graph learning, where each client trains a shared global model locally and uploads the trained model to the server. The server then aggregates all uploaded models to update the shared global model. We use $\mathcal{G} = (\mathcal{V}, \mathcal{E})$ to denote a graph, where $\mathcal{V}$ represents the set of nodes and $\mathcal{E} \subseteq \mathcal{V} \times \mathcal{V}$ represents the set of edges. Consider $K$ clients defined

as $\mathcal{C} = \{c_k\}_{k=1}^K$ manage disjoint graphs $\mathcal{G}_k = (\mathcal{V}_k, \mathcal{E}_k)$. We denote the global model for $t$-th round as $\mathcal{M}^t$ with parameters $w^t$, and the local model as $\mathcal{M}_k^t$ with corresponding parameters $w_k^t$. Each client $c_k$ possesses distinct private data $\mathcal{G}_k = (\mathcal{V}_k, \mathcal{E}_k)$. The global set of nodes is $\mathcal{V} = \bigcup_{k=1}^K \mathcal{V}_k$ with $\mathcal{V}_i \cap \mathcal{V}_j = \varnothing$ for all $i \neq j$. The adjacency matrix of $\mathcal{G}_k$ is defined as $\mathbf{A}_k = \{A_{ij}\}_{i,j \in \mathcal{V}_k}$, where $A_{ij} = 1$ if there is an edge between nodes $v_i$ and $v_j$, and $A_{ij} = 0$ otherwise. Similarly, $\mathbf{X}_k$ represents node features, and $\mathbf{Y}_k$ represents the corresponding label set.

**Client Homophily Level.** For a client $c_k$ with given graph $\mathcal{G}_k = (\mathcal{V}_k, \mathcal{E}_k)$ with nodes labeled by the $\mathbf{Y}_k$. The client homophily level is a measure of the consistency of connected nodes. It is defined as the ratio of edges that connect nodes with the same label to the total edges. Formally, the homophily level $h_k$ of client $c_k$ is calculated as:

$$h_k = \frac{\sum_{(i,j) \in \mathcal{E}_k} \mathbb{I}(\mathbf{y}_i = \mathbf{y}_j)}{|\mathcal{E}_k|}, \tag{1}$$

where $|\mathcal{E}_k|$ is the total count of edges in graph $\mathcal{G}_k$, and $\mathbb{I}(\mathbf{y}_i = \mathbf{y}_j)$ is a function that returns 1 if $y_i = y_j$ and 0 otherwise.

## 3.2 TOPOLOGY- AND DISTRIBUTION-AWARE CLIENT ESTIMATION (TDCE)

Unlike the traditional FL setting, the graph structure in the FGL setting provides more opportunities for trigger injection, making it harder to detect. Existing studies (Wan et al., 2025; Huang et al., 2024) primarily focus on utilizing data distribution to distinguish malicious clients, while neglecting the consideration of graph topological features. Moreover, the differences between homophily and heterophily edges further increase the difficulty of identifying trigger injections solely based on data distribution. Therefore, combining graph topological features and data distribution to detect trigger injections emerges as a promising solution. To this end, we propose the Topology- and Distribution-aware Client Estimation (TDCE) framework. TDCE synergistically combines energy-based modeling with topological feature analysis to provide a more holistic and discriminative client estimation, significantly enhancing the robustness of FGL against sophisticated backdoor attacks.

First, we extend the standard Energy-Based GCN (Wan et al., 2025) to incorporate both feature and structural information. A key feature of Energy-Based GCN is their flexibility: the energy function can be implemented using various forms of neural networks without strict structural constraints. For a given client $k$ with graph $\mathcal{G}_k = (\mathcal{V}_k, \mathcal{E}_k)$, we define the energy of a node $v_i$ as:

$$E_\theta(v_i) = -\log \sum_y \exp(f_\theta(\mathbf{x}_i, \mathcal{N}(v_i))[y]), \tag{2}$$

where $f_\theta(\cdot, \cdot)$ denotes the GCN forward pass that aggregates features from the node's neighborhood $\mathcal{N}(v_i)$. $f_\theta(\cdot, \cdot)[y]$ is the logits output of the model. This formulation captures both node features and local graph structure. The meta energy represents the unnormalized likelihood of the sample point. Lower energy corresponds to higher likelihood and consequently a greater probability of the sample being benign. To better the assignment of lower energy to benign samples and higher energy to malicious ones, we build upon prior studies (Hyvarinen & Dayan, 2005; Wan et al., 2025) by calculating the meta energy and deriving the energy shift score to construct the corresponding loss function. Specifically:

$$\mathcal{L}_{ESC} = \frac{1}{|\mathcal{V}_*|} \sum_{i=1}^{|\mathcal{V}_*|} \left[ \nabla_{v_i} S(v_i)^T \nabla_{v_i} S(v_i) + \frac{1}{2} \|S(v_i)\|^2 \right], \tag{3}$$

where $S(v_i)$ is the energy shift score between node energy $E_\theta(v_i)$ and perturbed meta energy $M_*(v_i)$. $M_*(v_i)$, which calculated by $E_\theta(\text{Perturb}(v_i))$, where $\text{Perturb}(v_i)$ conduct arbitrarily adding or removing edges connected to $v_i$ and perturbing both its features and those of its neighbors. Furthermore, the energy shift score between node energy $E_\theta(v_i)$ and perturbed meta energy $M_*(v_i)$ are calculated as:

$$S(v_i) = \left[ \frac{E_\theta(v_1) - M_1(v_i)}{E_\theta(v_1)}, \cdots, \frac{E_\theta(v_i) - M_N(v_i)}{E_\theta(v_i)} \right]. \tag{4}$$

Since the perturbation introduced by $\text{Perturb}(\cdot)$ involves both features and topology, there are theoretically infinite possibilities for $M_*(\cdot)$. We use $N$ to represent the number of instances involved

in the computation. A detailed analysis explaining why $\mathcal{L}_{ESC}$ improves the assignment of lower energy to benign samples and higher energy to malicious ones, along with the hyper-parameter sensitivity analysis for $N$, is provided in Appendix A.6.

To quantify the topological feature of each client's graph, we compute a comprehensive set of topological features $\mathbf{T}_k$ for each client $k$, including: Node Degree, Local Clustering Coefficient, Degree Centrality, PageRank Score, and Standard Deviation of Neighbor Degrees. Details for these features calculation can be found in Appendix A.1. These features are concatenated into a topological descriptor vector $\mathbf{T}_k$. The rich topological features further mitigate the potential noise introduced by structure-based perturbations in $\mathcal{L}_{ESC}$, ensuring the accuracy of graph topological features.

Then, we get a comprehensive topology- and distribution-aware estimation for each client $k$:

$$\mathbf{z}_k = \mathrm{MLP}_\phi\left(\left[\mathrm{norm}\left(\mathbf{E}_k\right); \mathrm{norm}\left(\mathbf{T}_k\right)\right]\right), \quad (5)$$

where $\mathbf{E}_k = \{E_\theta\left(v_i\right)\}_{v_i \in \mathcal{V}_k}$ is the energy distribution, $\mathrm{norm}(\cdot)$ denotes layer normalization, $[\cdot; \cdot]$ indicates concatenation, and $\mathrm{MLP}_\phi$ is a multi-layer perceptron that projects the combined vector into a latent space (To reduce computational costs, we utilize a single-layer MLP for practice).

## 3.3 TOPOLOGY- AND DISTRIBUTION-AWARE GRAPH CONSTRUCTION (TDGC)

To further enhance the discrimination between benign and malicious clients, we introduce a Topology- and Distribution-aware Graph Construction (TDGC) component. TDGC constructs a virtual graph $\bar{\mathcal{G}} = \left(\bar{\mathcal{V}}, \bar{\mathcal{E}}\right)$ at the server, where each node $\bar{v}_k \in \bar{\mathcal{V}}$ corresponds to a client $c_k$, and edges $\bar{\mathcal{E}}$ are established based on the similarity of the comprehensive client estimations $\mathbf{z}_k$ obtained from the TDCE component. This virtual graph enables the server to reason about the collective behavior of clients and identify potential malicious participants through graph structural analysis.

First, we define a similarity metric between two clients $c_m$ and $c_n$ that incorporates both their energy distributions and topological features, as encoded in $\mathbf{z}_m$ and $\mathbf{z}_n$. Specifically, we compute the cosine similarity between the client estimation vectors:

$$\mathcal{S}_{mn} = \frac{\mathbf{z}_m \cdot \mathbf{z}_n}{\|\mathbf{z}_m\| \|\mathbf{z}_n\|}, \quad (6)$$

where $\mathcal{S}_{mn} \in [-1, 1]$. To account for the homophily level $h_k$ of each client, we adjust the similarity measure as follows:

$$\tilde{\mathcal{S}}_{mn} = \mathcal{S}_{mn} \cdot \left(1 - |h_m - h_n|\right). \quad (7)$$

This adjustment penalizes similarity between clients with divergent homophily levels, as significant differences in homophily may indicate fundamentally different graph structures—potentially due to malicious perturbations.

We apply a threshold $\tau$ to binarize the similarities and construct the edge set $\bar{\mathcal{E}}$ of the virtual graph:

$$\bar{e}_{mn} = \begin{cases} 1, & \text{if } \tilde{\mathcal{S}}_{mn} \geq \tau \\ 0, & \text{otherwise} \end{cases}. \quad (8)$$

Here, $\bar{e}_{mn} = 1$ indicates that clients $c_m$ and $c_n$ are sufficiently similar in both distribution and topology, and an edge is established between them in $\bar{\mathcal{G}}$.

Notably, model for each clients are jointly optimized via cross entropy loss $\mathcal{L}_{CE}$ and energy shift loss $\mathcal{L}_{ESC}$ with equal weights.

## 3.4 SERVER-SIDE AGGREGATION

The virtual graph $\bar{\mathcal{G}}$ provides a structural representation of client relationships. We hypothesize that malicious clients, having injected anomalous triggers and altered local graph topologies, will exhibit estimation vectors that deviate from those of benign clients. Consequently, in $\bar{\mathcal{G}}$, malicious clients are expected to have fewer connections to benign ones.

We leverage this intuition to compute a maliciousness score $\Gamma_k$ for each client $c_k$ based on its degree centrality within $\bar{\mathcal{G}}$: $\Gamma_k = -\frac{d_k}{\max_{l \in \bar{\mathcal{V}}} d_l}$, where $d_k$ is the degree of node $\bar{v}_k$ in $\bar{\mathcal{G}}$. Clients with low

degrees are considered outliers and assigned higher maliciousness scores. This score is then used during aggregation to down-weight potentially malicious updates:

$$\zeta_k = \frac{\exp\left(-\gamma\Gamma_k\right)}{\sum_{l=1}^{K}\exp\left(-\gamma\Gamma_l\right)}, \tag{9}$$

where $\gamma > 0$ is a temperature parameter that controls the sharpness of the weight distribution. (For simplify, we utilize $\gamma = 1$ for practice).

For round $t$-th aggregation, the global model parameter are updated by: $w^{t+1} = \sum_{k}^{K} \zeta_k w_k^t$.

## 4 EXPERIMENT

To validate the effectiveness of FedTD, we conducted experiments under both IID and Non-IID-Louvain (Wang et al., 2022; Zhang et al., 2021a) settings on five datasets.

### 4.1 EXPERIMENT SETTINGS

**Datasets.** Following previous studies (Liu et al., 2023; Wan et al., 2025), we evaluate efficacy and robustness across three scenarios: Citation Network (Yang et al., 2016), Co-authorship (Shchur et al., 2018), and Amazon-Purchase (McAuley et al., 2015). Details can be found in Appendix A.3. Following previous studies (Dai et al., 2023; Wan et al., 2025), all original labeled datasets are divided into training, validation, and testing sets (60%/20%/20%). Unlabeled nodes are utilized for trigger injection and subsequently relabeled with the target class.

**Baselines.** We compare FedTD with various advanced baselines: 1) FedAvg (McMahan et al., 2017); 2) Trimmed Median and 3) Trimmed Mean (Yin et al., 2018); 4) FoolsGold (Fung et al., 2018); 5) DnC (Shejwalkar & Houmansadr, 2021); 6) SageFlow (Park et al., 2021); 7) MMA (Huang et al., 2023); 8) RLR (Ozdayi et al., 2021); 9) Freqfed (Fereidooni et al., 2023); 10) FedCPA (Han et al., 2023); 11) G$^2$uard (Yu et al., 2023); 12) FedGTA (Li et al., 2024a); 13) FGGP (Wan et al., 2024); 14) FedTGE (Wan et al., 2025). Detailed introduction of these methods can be found in Appendix A.4.

**Model Structure.** Following previous studies in FGL (Dai et al., 2023; Wan et al., 2025), we employ a two-layer GCN as both the feature extractor and classifier, using a hidden layer size of 32 across all datasets. For efficiency reasons, we fix $N = 1$.

**Backdoor Attack.** Following previous studies (Wan et al., 2025), we evaluate the robustness of FedTD under the widely adopt setting (Xu et al., 2021; Liu et al., 2023; Wan et al., 2025) (details in Appendix A.9). Given the stealthy nature of backdoor attacks, the trigger size is limited to 4 nodes in all experiments, with the trigger type set to Renyi and its location randomized. The malicious client ratio ($\Upsilon$) is configured as $\{0.1, 0.3, 0.5\}$, and experiments are conducted under both IID and Non-IID-Louvain settings. Results for $\Upsilon = 0.3$ are presented in Table 1 and Table 2, with additional results detailed in Appendix A.1.

**Metrics.** Following previous studies (Li et al., 2020; Liu et al., 2023; Wan et al., 2025), we adopt node classification accuracy $\mathcal{A}$ and backdoor failure rate $\mathcal{R}$ as the primary experimental metrics. Additionally, we use $\mathcal{B}$ as a comprehensive metric to evaluate the overall performance, considering both model accuracy and defense effectiveness. Notably, $\mathcal{B} = (\mathcal{A} + \mathcal{R})/2$.

**Implement Details.** To ensure the robustness and reliability of the results, each experiment is repeated five times for every federated method. We conduct a hyperparameter grid search for all baselines following the ranges suggested in their respective papers. The GNN models are trained using the Adam optimizer (Adam et al., 2014) with a learning rate of 0.01. For hyper-parameter, we conduct grid search for local training epoch $R$ from $\{5, 10, 15, 20\}$, and threshold $\tau$ from $\{0.4, 0.5, 0.6, 0.7\}$. Moreover, all trigger types (Renyi (Zhang et al., 2021b), GTA (Xi et al., 2021), WS (Watts & Strogatz, 1998), BA (Barabási & Albert, 1999), and Opt-GDBA (Yang et al., 2024)) for main experiment and additional experiments mentioned in Section 4.1 are introduced in Appendix A.5.

### 4.2 EXPERIMENT RESULTS

#### 4.2.1 PERFORMANCE COMPARISON

Table 1: Comparison with baselines over three distinct scenarios on five mainstream datasets under IID setting with a malicious proportion of $\Upsilon = 0.3$ and a trigger type of Renyi.

| Scenarios | Citation Network | | | | | | Co-authorship | | | | | | Amazon-purchase | | |
|---|---|---|---|---|---|---|---|---|---|---|---|---|---|---|---|
| Datasets | Cora | | | Pubmed | | | CS | | | Physics | | | Photo | | |
| Metrics | $\mathcal{A}$ | $\mathcal{R}$ | $\mathcal{B}$ | $\mathcal{A}$ | $\mathcal{R}$ | $\mathcal{B}$ | $\mathcal{A}$ | $\mathcal{R}$ | $\mathcal{B}$ | $\mathcal{A}$ | $\mathcal{R}$ | $\mathcal{B}$ | $\mathcal{A}$ | $\mathcal{R}$ | $\mathcal{B}$ |
| FedAvg | 74.25 | 19.47 | 46.86 | 86.12 | 4.26 | 45.19 | 85.90 | 20.47 | 53.19 | 93.51 | 16.37 | 54.94 | 81.61 | 5.33 | 43.47 |
| Trimmed Median | 73.67 | 30.93 | 52.30 | 85.84 | 7.61 | 46.73 | 86.01 | 7.87 | 46.94 | 93.46 | 17.25 | 55.52 | 82.47 | 6.32 | 44.40 |
| Trimmed Mean | 73.17 | 25.60 | 49.38 | 86.08 | 5.72 | 45.90 | 86.11 | 6.05 | 46.09 | 93.35 | 17.55 | 55.45 | 81.74 | 6.32 | 44.03 |
| FoolsGold | 77.25 | 33.87 | 55.56 | 87.33 | 13.44 | 50.39 | 85.69 | 10.60 | 48.15 | 93.33 | 23.94 | 58.63 | 84.82 | 5.02 | 44.92 |
| DnC | 66.25 | 76.00 | 71.13 | 86.05 | 24.09 | 55.07 | 85.89 | 66.12 | 76.01 | 93.17 | 41.35 | 67.26 | 71.76 | 16.36 | 44.06 |
| SageFlow | 75.08 | 39.47 | 57.28 | 87.17 | 6.13 | 46.65 | 86.03 | 14.50 | 52.76 | 94.17 | 23.32 | 58.75 | 84.92 | 25.97 | 53.28 |
| MMA | 75.00 | 36.27 | 55.63 | 86.98 | 6.67 | 46.83 | 87.04 | 13.77 | 50.41 | 93.87 | 22.16 | 58.02 | 85.55 | 23.81 | 54.68 |
| RLR | 76.00 | 13.07 | 44.53 | 86.77 | 14.78. | 50.78 | 84.02 | 15.37 | 49.70 | 93.56 | 14.53 | 54.05 | 78.26 | 6.84 | 42.55 |
| Freqfed | 76.25 | 18.67 | 47.56 | 86.16 | 7.25 | 46.71 | 86.93 | 10.93 | 48.93 | 93.38 | 6.78 | 50.08 | 81.76 | 4.33 | 43.05 |
| FedCPA | 74.42 | 24.27 | 49.34 | 85.60 | 8.77 | 47.19 | 86.87 | 15.12 | 51.00 | 93.99 | 19.32 | 56.66 | 80.76 | 4.33 | 42.55 |
| G²uard | 73.75 | 66.00 | 69.88 | 84.54 | 27.65 | 56.10 | 86.18 | 44.23 | 65.21 | 92.83 | 38.23 | 65.53 | 81.43 | 51.34 | 66.39 |
| FedGTA | 76.11 | 44.98 | 60.55 | 85.89 | 16.60 | 51.25 | 86.31 | 29.98 | 58.15 | 93.71 | 18.41 | 56.06 | 81.88 | 19.22 | 50.55 |
| FGGP | 77.02 | 32.54 | 54.78 | 86.29 | 14.10 | 50.20 | 85.91 | 22.85 | 54.38 | 93.49 | 18.20 | 55.85 | 81.59 | 8.98 | 45.29 |
| FedTGE | 75.00 | 70.47 | 72.83 | 85.79 | 57.19 | 71.49 | 85.63 | 70.45 | 78.04 | 93.99 | 57.02 | 75.51 | 80.81 | 97.92 | 89.37 |
| FedTD | 76.37 | 72.91 | **74.64** | 85.09 | 60.31 | **72.70** | 85.69 | 71.20 | **78.45** | 93.80 | 62.65 | **78.23** | 82.50 | 97.04 | **89.77** |

Table 2: Comparison with baselines over three distinct scenarios on five mainstream datasets under Non-IID-Louvain setting with a malicious proportion of $\Upsilon = 0.3$ and a trigger type of Renyi.

| Scenarios | Citation Network | | | | | | Co-authorship | | | | | | Amazon-purchase | | |
|---|---|---|---|---|---|---|---|---|---|---|---|---|---|---|---|
| Datasets | Cora | | | Pubmed | | | CS | | | Physics | | | Photo | | |
| Metrics | $\mathcal{A}$ | $\mathcal{R}$ | $\mathcal{B}$ | $\mathcal{A}$ | $\mathcal{R}$ | $\mathcal{B}$ | $\mathcal{A}$ | $\mathcal{R}$ | $\mathcal{B}$ | $\mathcal{A}$ | $\mathcal{R}$ | $\mathcal{B}$ | $\mathcal{A}$ | $\mathcal{R}$ | $\mathcal{B}$ |
| FedAvg | 61.81 | 25.24 | 45.53 | 85.81 | 5.85 | 45.83 | 90.57 | 39.87 | 65.22 | 94.61 | 39.58 | 67.09 | 71.98 | 66.88 | 69.43 |
| Trimmed Median | 63.85 | 22.15 | 43.00 | 86.01 | 17.85 | 51.93 | 87.68 | 50.84 | 69.26 | 94.52 | 45.96 | 70.24 | 63.45 | 76.13 | 69.79 |
| Trimmed Mean | 65.40 | 25.72 | 45.56 | 85.19 | 10.02 | 47.60 | 87.55 | 49.54 | 68.55 | 94.50 | 42.34 | 68.42 | 68.07 | 74.36 | 7121. |
| FoolsGold | 76.86 | 38.24 | 57.55 | 86.77 | 13.27 | 50.02 | 90.96 | 43.09 | 67.02 | 95.34 | 35.22 | 65.28 | 84.99 | 51.23 | 68.11 |
| DnC | 41.21 | 80.05 | 60.63 | 60.40 | 13.12 | 36.76 | 53.21 | 77.43 | 65.32 | 85.48 | 64.69 | 75.08 | 45.90 | 94.09 | 60.99 |
| SageFlow | 79.80 | 54.38 | 67.09 | 87.88 | 23.47 | 55.67 | 89.42 | 53.86 | 71.64 | 93.26 | 53.36 | 73.31 | 77.10 | 64.75 | 70.93 |
| MMA | 75.97 | 56.75 | 66.36 | 87.86 | 25.70 | 56.78 | 87.14 | 50.70 | 68.92 | 95.07 | 52.91 | 73.99 | 78.77 | 68.73 | 73.75 |
| RLR | 79.09 | 33.94 | 56.52 | 86.54 | 15.28 | 50.91 | 87.78 | 37.56 | 62.67 | 79.29 | 30.33 | 62.81 | 78.87 | 51.75 | 65.31 |
| Freqfed | 76.92 | 37.78 | 57.35 | 86.67 | 10.23 | 48.45 | 89.65 | 38.33 | 63.99 | 79.18 | 8.49 | 43.84 | 58.23 | 68.14 | 63.19 |
| FedCPA | 77.60 | 40.14 | 58.87 | 86.70 | 22.36 | 54.53 | 89.31 | 43.14 | 66.23 | 95.31 | 35.45 | 65.38 | 77.36 | 54.69 | 66.03 |
| G²uard | 73.66 | 47.82 | 60.74 | 83.34 | 33.47 | 58.41 | 88.15 | 55.89 | 72.02 | 93.38 | 41.87 | 67.63 | 72.34 | 67.29 | 69.82 |
| FedGTA | 76.12 | 37.92 | 57.02 | 86.03 | 19.64 | 52.84 | 89.19 | 48.32 | 68.76 | 93.98 | 42.98 | 68.48 | 73.59 | 70.79 | 72.19 |
| FGGP | 75.80 | 35.90 | 55.85 | 87.02 | 15.99 | 51.51 | 90.01 | 46.98 | 68.50 | 94.32 | 39.85 | 67.09 | 72.98 | 69.09 | 71.04 |
| FedTGE | 77.32 | 55.85 | 66.58 | 86.79 | 67.22 | 77.01 | 88.15 | 72.10 | 80.13 | 94.06 | 57.98 | 76.02 | 77.46 | 94.81 | 86.14 |
| FedTD | 78.78 | 58.02 | **68.40** | 87.21 | 69.07 | **78.14** | 89.92 | 73.89 | **81.91** | 95.09 | 61.09 | **78.09** | 77.91 | 94.69 | **86.30** |

Table 1 and Table 2 show the accuracy and defense performance of all baselines (including traditional FL defense methods and FGL methods) under both IID and Non-IID-Louvain settings[2]. The results demonstrate that FedTD exhibits outstanding performance under both IID and Non-IID-Louvain settings,

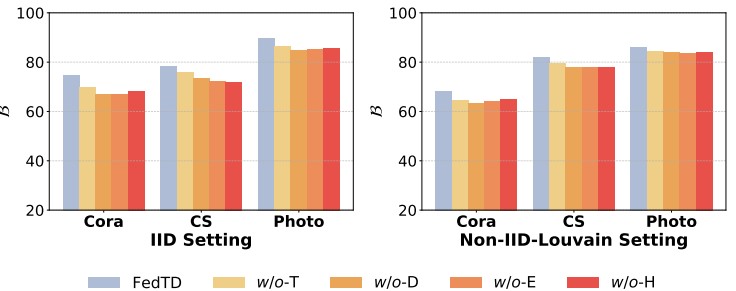

Figure 3: Ablation study under IID and Non-IID settings with Renyi.

proving its effectiveness against various attack patterns in FGL. Moreover, existing traditional FL defense methods perform poorly due to their lack of consideration for decentralization, while existing FGL methods underperform because they fail to defend against backdoor attacks. Although FedTGE attempts to defend against backdoor attacks from the perspective of the energy model, it still lacks sufficient consideration of topological features. FedTD better identifies backdoor attacks in FGL by simultaneously considering data distribution and topological features.

### 4.2.2 ABLATION STUDY

To analyze the effectiveness of FedTD, we conduct comprehensive ablation studies to evaluate the necessity and contribution of each individual component within FedTD. Specifically, we compare FedTD with following variants: 1) $w/o$-T, which removes the topological features in TDCE component. 2) $w/o$-D, which removes data distribution in TDCE component. 3) $w/o$-E, which removes

---

[2]Due to space limitations, we only present the results of Renyi with a client ratio of $\Upsilon = 0.3$ in Table 1 and Table 2. More experiments are provided in Appendix A.7.

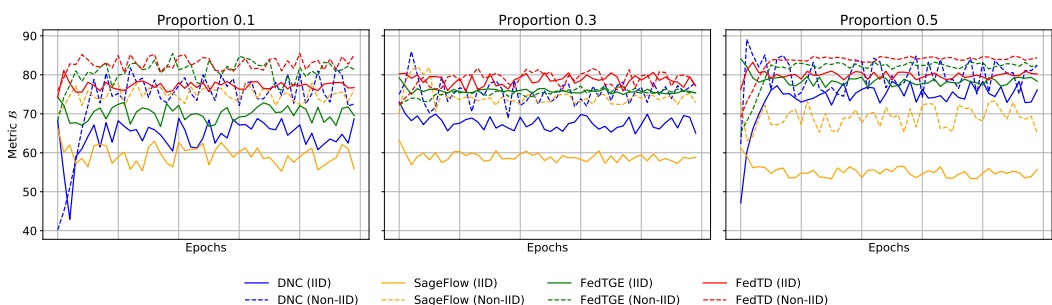

Figure 4: Training Curve with Renyi type trigger on Physics dataset.

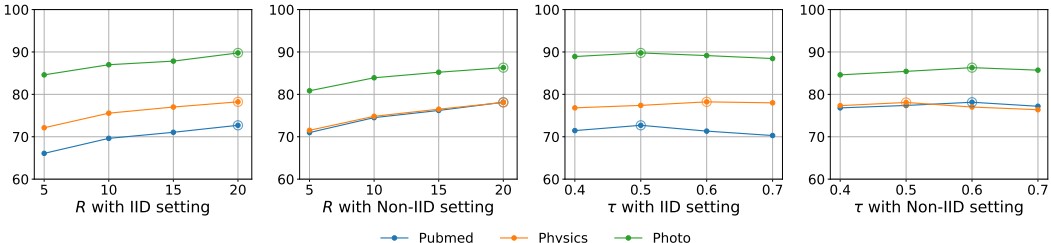

Figure 5: Hyper-parameter analysis under both IID and Non-IID settings with Renyi type trigger.

$\mathcal{L}_{ECS}$ in TDCE. 4) $w/o$-H, which removes adjustment about homophily level in TDGC component. We conduct experiments on Cora, CS, and Photo datasets to cover all three scenarios to ensure the reliability of the analysis. The evaluation results are demonstrated in Figure 3. We found that both $w/o$-T and $w/o$-D underperform compared to FedTD, which we attribute to the data distribution and topological features both significantly contribute to identifying triggers, highlighting the importance of considering topological features. The performance of $w/o$-E is inferior to FedTD, indicating that $\mathcal{L}_{ESC}$ contributes to better distinguishing between malicious and benign samples. Moreover, the performance degradation caused by $w/o$-H indicates that differences in homophily levels across clients affect trigger identification.

### 4.2.3 TRAINING CURVE

We plotted the curves of comprehensive metric $\mathcal{B}$ during the training curve on Physics dataset with various malicious proportions $\Upsilon = \{0.1, 0.3, 0.5\}$ in both IID and Non-IID-Louvain settings. We choose FedTGE, SageFlow, and Dnc as baselines due to their competitive performance. In Figure 4, we observe that FedTD demonstrates outstanding performance and exhibits a more stable curve across all settings.

### 4.2.4 HYPER-PARAMETER ANALYSIS

We evaluate the impact of the key hyper-parameters (local training epoch $R$ and threshold $\tau$) on FedTD's performance across three widely adopted datasets (Pubmed, Physics, and Photo) cover three distinct scenarios under both IID and Non-IID-Louvain settings. Figure 5 shows that a higher number of local training rounds $R$ benefits the model's performance, which aligns with the common experience in federated learning scenarios and can be dynamically adjusted based on computational resources. In contrast, the choice of threshold $\tau$ in FedTD appears less significant. Empirically, values of 0.5 or 0.6 tend to produce the most satisfactory results across all scenarios.

### 4.2.5 COMPLEXITY STUDY

In this section, we analyze the computational complexity of FedTD, which comprises two main components: TDCE and TDGC, examined separately.

**TDCE Complexity.** The TDCE component operates at the client level to compute both energy distributions and topological features. Let $D$ represent the number of nodes, $F$ the number of features

Table 3: Efficiency analysis for FedTD and baselines.

| Dataset | FedCPA | G$^2$uard | FedGTA | FGGP | FedTGE | FedTD |
|---|---|---|---|---|---|---|
| Pubmed | 32.16s | 38.29s | 37.21s | 44.76s | 26.29s | 28.05s |
| Physics | 102.27s | 119.22s | 115.57s | 135.00s | 82.93s | 84.29s |
| Photo | 44.19s | 51.20s | 49.14s | 60.09s | 37.29s | 39.02s |

per node, and $E$ the number of edges in the local graph. The computational complexity of TDCE consists of three main parts: energy distribution computation via Energy-based GCN, topological feature extraction, and the final estimation fusion.

For the energy distribution computation: Forward propagation through GCN: $\mathcal{O}(E \times F) + \mathcal{O}(D \times F)$. Energy shift score calculation with N perturbations: $\mathcal{O}(D \times N)$. Loss computation LESC: $\mathcal{O}(D)$.

For topological feature extraction: Node degree and centrality: $\mathcal{O}(D)$. Local clustering coefficient: $\mathcal{O}(D \times d^2)$, where $d$ is average degree. PageRank: $\mathcal{O}(E \times Iter)$, where $Iter$ is iteration count. Neighbor degree statistics: $\mathcal{O}(E)$.

Since $E$ is proportional to $D$ in non-dense graphs, and considering the MLP fusion step, the total TDCE complexity is:

$$\mathcal{O}(E \times F \times N) + \mathcal{O}(D \times d^2) \approx \mathcal{O}(D \times F \times N). \tag{10}$$

**TDGC Complexity.** The TDGC component operates at the server level to construct the virtual graph and compute maliciousness scores. Let $K$ denote the number of clients and $L$ the length of energy distributions.

The computational complexity of TDGC involves: Similarity matrix computation: $\mathcal{O}(K^2 \times L)$. Homophily adjustment: $\mathcal{O}(K^2)$. Virtual graph construction with threshold $\tau$: $\mathcal{O}(K^2)$. Maliciousness score calculation: $\mathcal{O}(K)$. Weight computation: $\mathcal{O}(K)$.

The overall TDGC complexity can be expressed as:

$$\mathcal{O}(K^2 \times L) + \mathcal{O}(K^2) + \mathcal{O}(K) \approx \mathcal{O}(K^2 \times L). \tag{11}$$

**Overall Complexity.** Since $K$ (number of clients) and $L$ (energy distribution length) are typically small constants in practical federated settings, and $N$ (perturbation count) is fixed at 1 for efficiency, the complexity simplifies to:

$$\mathcal{O}(D \times F) + \mathcal{O}(K^2). \tag{12}$$

This demonstrates that FedTD scales linearly with the graph size (nodes and features) and quadratically with the number of clients, making it suitable for large-scale graph datasets while maintaining reasonable computational overhead in typical federated scenarios with moderate client numbers.

Furthermore, we conducted empirical experiments to compare the efficiency of FedTD with advanced baselines. The experimental setup remains consistent with Table 2. As shown in Table 3, the results confirm that FedTD exhibits competitive efficiency, aligning with our aforementioned analysis.

Notably, The only additional communication introduced by FedTD is the upload of the client estimation vector $z_k$ from each client to the server. The dimension of $z_k$ is a hyperparameter (we used a small MLP); in our experiments, it was a low-dimensional vector. This constitutes a negligible increase in bandwidth compared to transmitting the entire local model update $w_k^t$.

## 5 CONCLUSION

In this paper, we propose FedTD, a tailored topology- and distribution-aware backdoor defense method for federated graph learning. At the client level, an energy function is introduced to integrate the underlying data distribution into the local model, assigning low energy to benign clients and high energy to malicious clients. By combining topological features with the energy function, FedTD achieves a more comprehensive and accurate energy estimation. At the server level, a virtual graph is constructed based on the estimation results from each client to evaluate their maliciousness scores.

To ensure the reliability of the virtual graph, the homophily level of each local graph is considered. During aggregation, clients with high maliciousness scores are assigned lower weights, while those with low scores are assigned higher weights, resulting in a more robust federated graph learning process. FedTD demonstrates strong robustness under both small and large malicious client ratios. Extensive experimental results across various federated graph scenarios under backdoor attacks validate the effectiveness of FedTD. By integrating graph topological features and data distribution, our work provides a comprehensive and effective solution for backdoor defense in federated graph learning, paving the way for future research in this direction.

## 6 Ethics statement

Our research complies with the ICLR Code of Ethics.

## 7 Reproducibility statement

The code can be found at the anonymous repository link provided at the end of the abstract. The appendix includes more comprehensive experimental results for different trigger types and malicious proportions.

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

# A APPENDIX

## A.1 TOPOLOGICAL FEATURES

In our Topology- and Distribution-aware Client Estimation (TDCE) component, we compute a set of five key topological features at the node level. These features are subsequently aggregated to form a client-level topological profile, which is used in conjunction with energy distributions to detect anomalous (potentially malicious) clients. The features were chosen for their ability to capture different aspects of a node's structural role within a graph, which is critical for identifying the subgraph patterns often introduced by backdoor triggers.

The features for each node $v_i$ in a client's graph $\mathcal{G}_k$ are calculated as follows:

### A.1.1 NODE DEGREE ($d_i$)

**Definition:**

The number of edges incident to a node. It is the most fundamental measure of a node's connectivity.

**Formula:**

$$d_i = \sum_{j=1}^{|\mathcal{V}_k|} A_{ij}, \tag{13}$$

where $A$ is the adjacency matrix of the graph and $|\mathcal{V}_k|$ is the total number of nodes.

**Relevance:**

Nodes with anomalously high or low degree in the context of their local graph may be part of an injected trigger structure.

### A.1.2 LOCAL CLUSTERING COEFFICIENT ($C_i$)

**Definition:**

Measures the degree to which a node's neighbors are connected to each other, indicating the tightness of its local clique structure.

**Formula:**

$$C_i = \frac{2\left|\{e_{jk} : v_j, v_k \in \mathcal{N}(v_i), e_{jk} \in \mathcal{E}_k\}\right|}{d_i(d_i - 1)}, \tag{14}$$

where $\mathcal{N}(v_i)$ is the set of neighbors of $v_i$ and $\mathcal{E}_k$ is the set of edges. The denominator represents the maximum possible number of edges between the neighbors.

**Relevance:**

Triggers often have significantly different structure from the original graph.

### A.1.3 DEGREE CENTRALITY ($C_D(v_i)$)

**Definition:**

A normalized version of node degree that allows for comparison across graphs of different sizes. It measures a node's immediate network influence.

**Formula:**

$$C_D\left(v_i\right) = \frac{d_i}{|\mathcal{V}_k| - 1}. \tag{15}$$

**Relevance:**

This feature helps identify nodes that are anomalously central within their local subgraph, which could be a characteristic of a trigger 'hub'.

### A.1.4 PAGERANK SCORE ($PR(v_i)$))

**Definition:**

An algorithm that measures the transitive influence or importance of a node based on the quantity and quality of its connections. A node has high PageRank if it is linked to by other important nodes.

**Formula:**

The PageRank score is computed iteratively until convergence:

$$PR\left(v_i\right) = \frac{1 - \alpha}{|\mathcal{V}_k|} + \alpha \sum_{v_j \in \mathcal{N}(v_i)} \frac{PR\left(v_j\right)}{d_j}, \tag{16}$$

where $\alpha \in [0, 1]$ is a damping factor (typically set to 0.85), representing the probability of a random surfer following links.

**Relevance:**

PageRank helps identify nodes that are structurally important beyond their immediate connections. Malicious triggers may create anomalous importance patterns.

### A.1.5 STANDARD DEVIATION OF NEIGHBOR DEGREES ($\sigma_{d_{\mathcal{N}(v_i)}}$)

**Definition:**

Quantifies the dispersion or heterogeneity in the connectivity of a node's neighbors.

**Formula:**

Let the multiset of degrees for the neighbors of $v_i$ be $D_{\mathcal{N}(v_i)} = d_j : v_j \in \mathcal{N}(v_i)$. Then:

$$
\begin{aligned}
\mu_{\mathcal{N}(v_i)} &= \frac{1}{d_i} \sum_{v_j \in \mathcal{N}(v_i)} d_j \\
\sigma_{d_{\mathcal{N}(v_i)}} &= \sqrt{\frac{1}{d_i} \sum_{v_j \in \mathcal{N}(v_i)} \left(d_j - \mu_{\mathcal{N}(v_i)}\right)^2}.
\end{aligned}
\tag{17}
$$

**Relevance:**

This is a crucial feature for detecting nodes that act as 'bridges' between regions of the graph with vastly different connectivity patterns—a role that nodes in a trigger subgraph may be forced into.

## A.2 ALGORITHM

## A.3 DATASETS

In this part, we introduce all datasets in details:

---

**Algorithm 1** Process of FedTD

---

1: **Input:** Total Rounds $T$, clients number $K$, global model $\mathcal{M}$ with corresponding parameter $w$, local models for all clients $\{\mathcal{M}_k\}_{k=1}^K$, with corresponding parameters $\{w_k\}_{k=1}^K$, and graphs for all clients $\{\mathcal{G}_k\}_{k=1}^K$;
2: **Output:** The final optimized global model $\mathcal{M}^T$;
3: **for** $t = 1$ to $T$ **do**
4:    *Client-Side:*
5:    **for** $k = 1$ to $K$ **in parallel do**
6:       $\mathcal{M}_k^t \leftarrow \text{LocalUpdate}(w_k^t, \mathcal{G}_k)$; // Utilizing Original Training Strategy.
7:       $\mathbf{E}_k \leftarrow \text{EnergyDistribution}(f_\theta(\cdot, \cdot), \mathbf{X}_k, \mathbf{Y}_k, \mathcal{G}_k)$; // Calculating Energy Distribution.
8:       $\mathbf{T}_k \leftarrow \text{TopologicalFearure}(\mathcal{G}_k)$; // Calculating Topological Features.
9:       $\mathbf{z}_k \leftarrow \text{Estimation}(\mathbf{E}_k, \mathbf{T}_k)$; // Calculating Final Estimation.
10:    **end for**
11:    *Server-Side:*
12:    $\tilde{\mathcal{S}} \leftarrow \text{SimilarityMatrix}(\{\mathbf{z}_k\}_{k=1}^K, \{h_k\}_{k=1}^K)$; // Calculating Similarity Matrix.
13:    $\bar{\mathcal{G}} \leftarrow \text{VirtualGraph}(\tilde{\mathcal{S}}, \tau)$; // Constructing Virtual Graph.
14:    $\{\Gamma_k\}_{k=1}^K \leftarrow \text{MaliciousnessScore}(\bar{\mathcal{G}})$; // Calculating Maliciousness Score.
15:    $\{\zeta_k\}_{k=1}^K \leftarrow \text{ClientWeight}(\{\Gamma_k\}_{k=1}^K)$; // Calculating Client Weights.
16:    $\mathcal{M}^{t+1} \leftarrow \text{GlobalUpdate}(\{\mathcal{M}_k^t\}_{k=1}^K, \{\zeta_k\}_{k=1}^K)$; // Updating Global Model.
17: **end for**
18: **return** $\mathcal{M}^T$;

---

Table 4: Statistics of the five evaluation datasets in three scenarios.

| Scenarios | Datasets | #Nodes | #Edges | #Classes | #Features |
|-----------|----------|--------|--------|----------|-----------|
| Citation Network | Cora | 2,708 | 5,278 | 7 | 1,433 |
| | Pubmed | 19,717 | 44,324 | 3 | 500 |
| Co-authorship | CS | 18,333 | 327,576 | 15 | 6,805 |
| | Physics | 34,493 | 495,924 | 5 | 8,415 |
| Amazon-purchase | Photo | 7,650 | 287,326 | 8 | 745 |

- **Citation Network:** Citation network datasets, including **Cora** and **PubMed**, comprise inter-connected research papers, where nodes represent individual studies and edges signify citation relationships. These datasets are widely used for tasks like research paper classification and knowledge graph construction, offering valuable insights into research trends and prominent topics within academic fields.

- **Co-authorship:** Co-authorship datasets, including **Computer Science (CS)** and **Physics**, are constructed from the Microsoft Academic Graph. In these datasets, nodes represent authors, while edges denote co-author relationships. They are commonly used to predict authors' research fields, facilitating the analysis of research collaboration networks, the distribution of research areas, and academic influence.

- **Amazon-purchase:** The Amazon-purchase dataset, including the **Photo** dataset, are derived from Amazon's co-purchase relationships. In these datasets, nodes represent products, and edges indicate co-purchase links. The primary goal is to predict product categories, making these datasets valuable for recommendation systems and market analysis research.

The statistics of the datasets used in our experiments are provided in Table **??**.

A.4   BASELINES

In this part, we introduce all baselines in details:

- **FedAvg:** A standard Federated Averaging algorithm aggregates updates from all participating clients at the server without implementing any defense strategies. Although it is a commonly used method in federated learning, it is notably vulnerable to adversarial attacks, making it a less secure option in hostile environments.

- **Trimmed Median** and **Trimmed Mean:** These methods aim to reduce the influence of malicious clients by filtering out outliers or abnormal parameter updates. Specifically, the trimmed median removes extreme values from client updates, while the trimmed mean discards a certain percentage of the largest and smallest values before performing aggregation. By doing so, these methods provide a level of defense against Byzantine failures and improve the robustness of the system.

- **FoolsGold:** A defense strategy is designed to counter model poisoning attacks by reducing the aggregation weight of clients with highly similar updates. The underlying assumption is that malicious clients often submit similar updates to amplify the effect of a backdoor attack. By assigning lower weights to these clients during aggregation, the method mitigates their impact and enhances the robustness of the model.

- **DnC:** This method groups clients into distinct clusters based on the similarity of their updates, followed by aggregating updates within each cluster. By isolating clients into separate clusters, DnC minimizes the risk of adversarial clients dominating the aggregation process, thereby making it more difficult for attackers to compromise the global model.

- **SageFlow:** A distance-based defense method identifies and mitigates malicious updates by analyzing discrepancies between client updates. Updates that deviate significantly from the majority are either discarded or assigned lower aggregation weights. This method enhances robustness by reducing the influence of adversarial clients on the global model.

- **MMA:** An adaptive defense method utilizes multiple metrics, including gradient norms, update similarities, and client performance, to detect and mitigate adversarial behaviors. By dynamically adjusting the aggregation weights of clients based on their performance across these metrics, this approach ensures a more robust and flexible defense against attacks.

- **RLR:** An adaptive learning rate defense method adjusts the learning rates of clients based on the perceived reliability of their updates. Clients submitting suspicious or noisy updates are assigned lower learning rates, reducing their influence on the global model. This strategy mitigates the impact of malicious clients and promotes stability during training.

- **Freqfed:** A frequency analysis-based defense method designed to mitigate poisoning attacks in federated learning. It leverages frequency domain transformations, such as Fourier transforms, to analyze client updates and identify malicious updates with anomalous frequency patterns. By decomposing updates into their frequency components, FreqFed effectively distinguishes between benign and adversarial modifications. This method ensures that only updates with consistent and expected frequency characteristics are aggregated, thereby reducing the impact of poisoning attacks and enhancing the robustness and integrity of the global model.

- **FedCPA:** A resilient method designed to counter malicious client updates through critical parameter analysis. It evaluates the significance of each parameter in the global model and selectively aggregates updates, focusing on the most impactful parameters. By prioritizing these critical parameters, FedCPA minimizes the attack surface available to adversaries, thereby strengthening the overall robustness of the learning process. This strategy ensures that even if certain clients are compromised, their ability to disrupt or manipulate the global model remains limited. Ultimately, FedCPA preserves the integrity and performance of the model, providing a secure and attack-tolerant federated learning solution.

- **G$^2$uard:** A defense method that protects federated learning systems from backdoor attacks by utilizing attributed client graph clustering. It reimagines the process of identifying malicious clients as an attributed graph clustering problem, applying clustering methods to detect adversarial participants. By introducing adaptive mechanisms, G2uard amplifies the differences between aggregated models and compromised ones, effectively neutralizing backdoor threats. Theoretical analysis confirms that G2uard preserves the convergence of the federated learning system, while empirical results demonstrate its effectiveness in significantly reducing attack success rates across various scenarios, with minimal impact on the model's performance on benign data.

- **FedGTA:** A novel Federated Graph Topology-aware Aggregation to address the challenges in FGL. Unlike conventional methods, FedGTA integrates topology-aware local smoothing confidence and

mixed moments of neighbor features to enhance model aggregation and performance. By encoding graph topology and node attributes, FedGTA customizes aggregation strategies for each client, ensuring scalability and efficiency in handling large-scale graphs. Extensive experiments on 12 real-world datasets demonstrate that FedGTA achieves state-of-the-art performance, bridging the gap between large-scale graph learning and FGL with superior generalization and robustness.

- **FGGP:** An innovative method designed to tackle the challenges of domain shifts in FGL. FGGP decouples the global model into a feature extractor and a classification model, connected by prototypes that serve as semantic centers. At the classification level, it employs Federated Cluster Prototypes Prediction (FCPP) to retain domain signals and enhance multi-domain prediction. At the feature extractor level, it proposes Global Knowledge Injected Contrast (GKIC), which leverages contrastive learning to align local data with global knowledge, enriching feature diversity and improving prototype quality. Extensive experiments demonstrate FGGP's effectiveness in significantly enhancing multi-domain generalization while addressing attribute and structure shifts, setting a new benchmark for FGL in real-world heterogeneous environments.

- **FedTGE:** An innovative method for defending FGL against backdoor attacks, which exploit the unique topological and heterogeneous nature of graph data. The proposed method, FedTGE, leverages energy-based modeling to enhance defense mechanisms. At the client level, it injects structural energy awareness into local models, assigning lower energy to benign samples and higher energy to malicious ones. At the server level, FedTGE clusters clients based on their energy distributions and constructs a global energy graph to propagate energy similarity, adjusting aggregation weights to prioritize benign clients. Extensive experiments on multiple datasets under IID and Non-IID settings demonstrate that FedTGE robustly mitigates backdoor attacks, outperforming state-of-the-art methods while maintaining high accuracy and scalability in diverse FGL scenarios.

## A.5 TRIGGER TYPE

In this part, we introduce all trigger types below:

- **Renyi:** The Renyi trigger, based on the Erdős–Rényi random graph model, introduces random nodes and edges into the graph. Each edge is formed with an independent probability, creating a structure devoid of discernible patterns. This high level of randomness provides strong obfuscation, making the trigger challenging to detect.

- **GTA:** The GTA trigger utilizes carefully designed, structured subgraphs (e.g., star or ring shapes) that are injected into the graph. By simultaneously altering the features of the inserted nodes, the attacker enhances their influence on targeted classifications. Although this design achieves high attack success rates, its structural regularity increases the risk of detection by defense methods.

- **WS:** The WS trigger, based on the Watts-Strogatz small-world model, introduces subgraphs with high clustering coefficients and short average path lengths, mimicking the characteristics of real-world networks to enhance stealthiness. The high local clustering significantly impacts the graph's global properties, thereby increasing attack efficacy. However, the success of the attack heavily relies on carefully selecting parameters such as rewiring probabilities.

- **BA:** The BA trigger, based on the Barabási-Albert scale-free network model, creates subgraphs with a power-law degree distribution, where a few 'hub' nodes possess high connectivity. These hubs effectively amplify the backdoor effect across the graph, leveraging their connectivity for more potent attacks. Although this structure closely resembles real-world networks, the addition of highly connected hub nodes may necessitate significant graph modifications, potentially affecting efficiency.

- **Opt-GDBA:** Opt-GDBA trigger is a learned subgraph that is adaptively generated for each individual graph using an adaptive trigger generator. This generator optimizes both the location and shape of the trigger by leveraging graph structure and node features, ensuring the trigger is attached to the most influential nodes. This results in highly effective and stealthy attacks.

## A.6 ANALYSIS FOR ENERGY SHIFT COMPUTATION LOSS

Previous studies (Deng et al., 2020; Wu et al., 2023; Wan et al., 2025) have shown that directly maximizing the density function of an energy-based model faces a significant computational burden

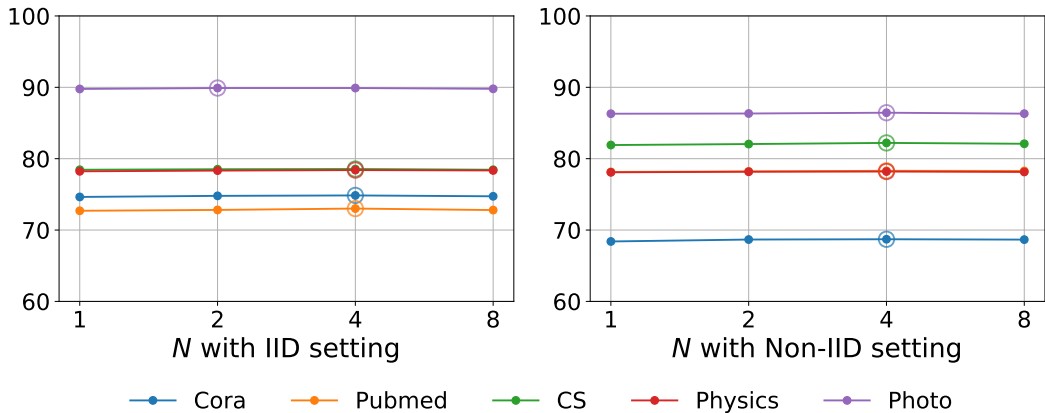

Figure 6: Analysis for $N$ under both IID and Non-IID settings with Renyi type trigger.

due to the normalization partition function. Therefore, using score matching has emerged as a promising alternative. Score matching is a technique for training energy-based models by aligning the gradient of the log-probability density function. By transforming the distribution into an equivalent score, we can train energy-based models more efficiently. Score matching, through $\nabla_x \log p_\theta(x) = -\nabla_x E_\theta(x)$, eliminates the need for a normalization constant in the density function. Therefore, Eq. 4 is equivalent to the following equation:

$$S\left(v_i\right) = \left[\frac{\log p_\theta\left(v_1\right) - \log p_\theta\left(\text{Perturb}(v_1)_1\right)}{\log p_\theta\left(v_1\right)}, \cdots, \frac{\log p_\theta\left(v_1\right) - \log p_\theta\left(\text{Perturb}(v_1)_N\right)}{\log p_\theta\left(v_1\right)}\right].$$

(18)

This implies that using the gradient surrogate $\nabla E_\theta$ enables the model to learn the energy density distribution of the real data $p_{\text{data}}\left(v_i\right)$. With an effective score proxy for the gradient in place, we continue to adhere to the traditional score matching objective:

$$D_F\left(p_{\text{data}}(\mathbf{x})\|p_\theta(\mathbf{x})\right) = \mathbb{E}_{p_{\text{data}}(\mathbf{x})}\left[\frac{1}{2}\left\|\nabla_\mathbf{x} \log p_{\text{data}}(\mathbf{x}) - \nabla_\mathbf{x} \log p_\theta(\mathbf{x})\right\|^2\right].$$

(19)

With the energy score surrogate (Eq. 4 equal to Eq. 18), our optimization objective is formulated as:

$$D_F\left(p_{\text{data}}\left(v_i\right)\|p_\theta\left(v_i\right)\right) = \mathbb{E}_{p_{\text{data}}(v_i)}\left[\frac{1}{2}\left\|S_{\text{data}}\left(v_i\right) - S\left(v_i\right)\right\|^2\right].$$

(20)

Since $S_{\text{data}}$ of the real data is unknown during the actual training of the model. Following previous study (Hyvarinen & Dayan, 2005) and incorporating my gradient proxy while reducing computational complexity, we rewrite it as Eq. 3. Energy shift computation loss $\mathcal{L}_{ECS}$ in Eq. 3 effectively increase $M_*(v_i)/E_\theta(v_i)$. This perfectly aligns with our goal of assigning lower energy to benign samples and higher energy to malicious ones.

In Section 4, we fixed $N$ to 1 for efficiency considerations. Here, we further perform a sensitivity analysis on the hyperparameter $N$. Specifically, we conduct experiments on five datasets under both IID and Non-IID settings with Renyi type triggers, varying $N$ within the range of $\{1, 2, 4, 8\}$. The experimental results in Figure 6 show that increasing $N$ can lead to performance improvements. We attribute this to the fact that more computational samples reduce the loss or misrepresentation of topological features caused by random perturbations when generating $M_*(v_i)$. However, in TCDE, $\mathbf{T}$ already provides sufficiently accurate topological features. Therefore, the performance improvement brought by increasing $N$ is not significant. For efficiency considerations, the performance with $N = 1$ is already satisfactory.

## A.7 ADDITIONAL EXPERIMENTS

In this part, we provide comprehensive experiments to validate the effectiveness and reliability of FedTD, covering various malicious client ratio and trigger types. Experiments results are demonstrated in Table 5, Table 1, Table 6, Table 7, Table 2, and Table 8, show the superiority of FedTD under different malicious client ratio. Moreover, Experiments results are demonstrated in Table 1,

Table 5: Comparison with baselines over three distinct scenarios on five mainstream datasets under IID setting with a malicious proportion of $\Upsilon = 0.1$ and a trigger type of Renyi.

| Scenarios | Citation Network | | | | | | Co-authorship | | | | | | Amazon-purchase | | |
|---|---|---|---|---|---|---|---|---|---|---|---|---|---|---|---|
| Datasets | Cora | | | Pubmed | | | CS | | | Physics | | | Photo | | |
| Metrics | $\mathcal{A}$ | $\mathcal{R}$ | $\mathcal{B}$ | $\mathcal{A}$ | $\mathcal{R}$ | $\mathcal{B}$ | $\mathcal{A}$ | $\mathcal{R}$ | $\mathcal{B}$ | $\mathcal{A}$ | $\mathcal{R}$ | $\mathcal{B}$ | $\mathcal{A}$ | $\mathcal{R}$ | $\mathcal{B}$ |
| FedAvg | 74.08 | 44.80 | 59.44 | 85.81 | 7.21 | 46.51 | 90.21 | 18.58 | 54.39 | 93.89 | 18.84 | 85.37 | 90.68 | 17.92 | 54.30 |
| Trimmed Median | 69.33 | 53.60 | 61.47 | 85.63 | 6.60 | 46.12 | 90.05 | 19.45 | 54.75 | 93.71 | 21.16 | 57.43 | 90.26 | 20.26 | 55.26 |
| Trimmed Mean | 73.25 | 36.80 | 55.03 | 85.77 | 6.60 | 46.18 | 90.15 | 17.38 | 53.77 | 93.84 | 24.29 | 59.06 | 90.42 | 19.74 | 55.08 |
| FoolsGold | 76.17 | 44.00 | 60.08 | 87.42 | 8.63 | 48.03 | 90.89 | 19.13 | 55.01 | 94.27 | 21.10 | 57.69 | 91.39 | 16.62 | 54.01 |
| DnC | 64.50 | 80.80 | 72.65 | 84.38 | 8.12 | 46.25 | 83.44 | 36.97 | 60.21 | 92.67 | 46.78 | 69.73 | 88.26 | 9.61 | 48.94 |
| SageFlow | 76.08 | 60.80 | 68.44 | 87.04 | 9.54 | 48.29 | 90.09 | 22.62 | 56.36 | 94.14 | 25.51 | 59.82 | 91.50 | 47.01 | 69.26 |
| MMA | 75.33 | 63.20 | 69.27 | 86.93 | 8.32 | 47.63 | 90.84 | 20.11 | 55.48 | 94.15 | 19.83 | 56.99 | 91.39 | 48.31 | 69.85 |
| RLR | 75.58 | 72.00 | 73.79 | 87.29 | 10.05 | 48.67 | 91.81 | 18.91 | 55.36 | 94.54 | 15.13 | 54.84 | 91.24 | 20.26 | 55.75 |
| Freqfed | 75.00 | 71.99 | 73.50 | 86.85 | 9.14 | 47.99 | 90.60 | 18.58 | 54.59 | 93.57 | 6.78 | 50.18 | 90.45 | 16.88 | 53.67 |
| FedCPA | 74.17 | 64.00 | 49.08 | 85.29 | 9.14 | 47.22 | 89.98 | 20.77 | 55.37 | 93.20 | 24.41 | 58.80 | 90.66 | 24.68 | 57.67 |
| G²uard | 74.17 | 48.00 | 61.09 | 84.69 | 34.87 | 59.78 | 89.63 | 33.46 | 61.55 | 93.44 | 34.69 | 64.07 | 89.34 | 79.61 | 84.48 |
| FedGTA | 75.61 | 62.12 | 68.87 | 85.09 | 18.42 | 51.76 | 90.82 | 24.09 | 57.46 | 93.28 | 29.35 | 61.32 | 90.99 | 37.12 | 64.06 |
| FGGP | 75.93 | 58.92 | 67.43 | 85.70 | 20.02 | 52.86 | 90.95 | 23.80 | 57.38 | 91.05 | 33.09 | 62.07 | 90.52 | 42.09 | 66.31 |
| FedTGE | 75.42 | 84.00 | _79.71_ | 85.12 | 46.19 | _65.66_ | 90.31 | 40.43 | _65.37_ | 93.28 | 57.10 | _70.42_ | 90.76 | 98.44 | **94.60** |
| FedTD | 75.81 | 85.02 | **80.42** | 85.61 | 48.22 | **66.92** | 90.98 | 41.88 | **66.43** | 94.02 | 58.62 | **76.32** | 91.05 | 96.58 | _93.82_ |

Table 6: Comparison with baselines over three distinct scenarios on five mainstream datasets under IID setting with a malicious proportion of $\Upsilon = 0.5$ and a trigger type of Renyi.

| Scenarios | Citation Network | | | | | | Co-authorship | | | | | | Amazon-purchase | | |
|---|---|---|---|---|---|---|---|---|---|---|---|---|---|---|---|
| Datasets | Cora | | | Pubmed | | | CS | | | Physics | | | Photo | | |
| Metrics | $\mathcal{A}$ | $\mathcal{R}$ | $\mathcal{B}$ | $\mathcal{A}$ | $\mathcal{R}$ | $\mathcal{B}$ | $\mathcal{A}$ | $\mathcal{R}$ | $\mathcal{B}$ | $\mathcal{A}$ | $\mathcal{R}$ | $\mathcal{B}$ | $\mathcal{A}$ | $\mathcal{R}$ | $\mathcal{B}$ |
| FedAvg | 72.00 | 15.36 | 43.68 | 85.91 | 5.97 | 45.94 | 90.09 | 12.48 | 51.28 | 93.88 | 16.33 | 55.11 | 90.53 | 20.36 | 55.45 |
| Trimmed Median | 69.17 | 11.68 | 40.42 | 85.58 | 5.02 | 45.30 | 90.37 | 9.36 | 49.86 | 93.46 | 11.85 | 52.65 | 89.71 | 18.29 | 54.00 |
| Trimmed Mean | 70.75 | 18.72 | 44.74 | 85.98 | 5.44 | 45.71 | 90.03 | 10.71 | 50.37 | 93.83 | 14.98 | 54.41 | 90.45 | 21.25 | 55.85 |
| FoolsGold | 73.83 | 15.52 | 44.68 | 87.17 | 5.56 | 46.37 | 90.87 | 11.89 | 51.38 | 94.19 | 15.01 | 54.60 | 91.13 | 10.13 | 50.63 |
| DnC | 62.33 | 55.04 | 58.69 | 83.77 | 4.65 | 44.21 | 87.61 | 82.19 | 84.75 | 92.65 | 57.45 | 75.05 | 87.29 | 20.68 | 53.98 |
| SageFlow | 75.58 | 26.24 | 50.91 | 87.14 | 5.28 | 46.21 | 90.55 | 5.46 | 48.01 | 92.99 | 17.33 | 55.16 | 91.18 | 17.25 | 54.22 |
| MMA | 75.33 | 15.36 | 45.35 | 87.39 | 5.42 | 46.41 | 90.98 | 11.26 | 51.12 | 94.18 | 18.52 | 56.35 | 91.08 | 20.42 | 55.75 |
| RLR | 76.33 | 12.48 | 44.41 | 87.28 | 5.50 | 46.39 | 91.56 | 11.93 | 51.75 | 94.53 | 11.61 | 53.07 | 91.16 | 11.48 | 51.32 |
| Freqfed | 74.05 | 28.61 | 51.33 | 85.21 | 5.28 | 45.24 | 90.77 | 9.29 | 50.03 | 93.05 | 11.88 | 52.56 | 90.05 | 15.06 | 52.59 |
| FedCPA | 74.17 | 11.20 | 42.68 | 86.01 | 5.18 | 45.59 | 90.60 | 10.93 | 50.77 | 93.37 | 13.22 | 53.30 | 90.05 | 9.61 | 49.83 |
| G²uard | 69.17 | 45.60 | 57.38 | 83.47 | 28.58 | 56.03 | 88.74 | 65.79 | 77.27 | 93.61 | 52.89 | 73.26 | 89.92 | 64.56 | 77.24 |
| FedGTA | 68.99 | 29.19 | 49.09 | 80.31 | 19.92 | 50.12 | 82.24 | 20.83 | 51.54 | 85.91 | 22.05 | 53.98 | 82.55 | 28.72 | 55.64 |
| FGGP | 66.87 | 33.02 | 49.95 | 80.04 | 22.34 | 51.19 | 81.67 | 24.15 | 52.91 | 86.50 | 23.19 | 54.85 | 82.49 | 35.72 | 59.11 |
| FedTGE | 72.80 | 73.76 | _73.28_ | 85.47 | 53.56 | _69.52_ | 90.21 | 80.63 | _85.42_ | 93.69 | 62.52 | _78.10_ | 90.55 | 92.62 | _91.59_ |
| FedTD | 74.95 | 75.03 | **74.99** | 87.09 | 58.90 | **73.00** | 90.80 | 80.52 | **85.66** | 93.70 | 66.01 | **79.86** | 91.09 | 93.00 | **92.05** |

Table 9, Table 10, Table 11, Table 12, Table 2, Table 13, Table 14, Table 15 and Table 16, show the superiority of FedTD under different trigger types. It is worth noting that for Opt-GDBA, which considers node importance and graph structure when designing triggers, our FedTD demonstrates significantly greater advantages compared to other baselines.

## A.8 DIFFERENT GRAPH BACKBONES

To demonstrate the generality of the proposed FedTD method across different graph architectures, we further examined the impact of replacing GCN with GAT (Velivckovic et al., 2018) and GraphSage (Hamilton et al., 2017) on FedTD's performance. Consistent with prior work (Liu et al., 2023), we utilized two-layer GAT and GraphSage models for this analysis. The results in Table 17 show that

Table 7: Comparison with baselines over three distinct scenarios on five mainstream datasets under Non-IID-Louvain setting with a malicious proportion of $\Upsilon = 0.1$ and a trigger type of Renyi.

| Scenarios | Citation Network | | | | | | Co-authorship | | | | | | Amazon-purchase | | |
|---|---|---|---|---|---|---|---|---|---|---|---|---|---|---|---|
| Datasets | Cora | | | Pubmed | | | CS | | | Physics | | | Photo | | |
| Metrics | $\mathcal{A}$ | $\mathcal{R}$ | $\mathcal{B}$ | $\mathcal{A}$ | $\mathcal{R}$ | $\mathcal{B}$ | $\mathcal{A}$ | $\mathcal{R}$ | $\mathcal{B}$ | $\mathcal{A}$ | $\mathcal{R}$ | $\mathcal{B}$ | $\mathcal{A}$ | $\mathcal{R}$ | $\mathcal{B}$ |
| FedAvg | 62.52 | 40.69 | 51.60 | 85.41 | 22.19 | 53.80 | 90.08 | 35.30 | 62.69 | 92.54 | 46.91 | 69.72 | 74.44 | 70.79 | 72.61 |
| Trimmed Median | 65.07 | 46.90 | 55.98 | 86.23 | 21.09 | 53.66 | 90.15 | 41.21 | 65.68 | 93.29 | 53.01 | 73.15 | 66.36 | 65.26 | 65.81 |
| Trimmed Mean | 65.12 | 44.83 | 54.97 | 86.20 | 20.50 | 53.35 | 90.18 | 38.36 | 64.27 | 93.83 | 50.20 | 72.02 | 69.70 | 67.63 | 68.67 |
| FoolsGold | 78.84 | 62.07 | 70.45 | 87.86 | 25.57 | 56.77 | 91.20 | 46.90 | 69.05 | 95.44 | 51.09 | 73.26 | 85.24 | 62.89 | 74.07 |
| DnC | 41.94 | 100.00 | 70.97 | 57.57 | 90.95 | 74.56 | 56.50 | 67.40 | 61.95 | 86.01 | 71.13 | 78.57 | 44.06 | 95.53 | 69.80 |
| SageFlow | 76.13 | 60.53 | 68.33 | 87.76 | 26.77 | 57.26 | 86.30 | 61.28 | 73.79 | 95.14 | 55.00 | 75.07 | 76.13 | 66.84 | 71.49 |
| MMA | 77.49 | 60.26 | 68.88 | 87.60 | 25.27 | 56.44 | 86.72 | 66.62 | 76.66 | 94.88 | 57.42 | 76.15 | 80.20 | 60.26 | 70.23 |
| RLR | 80.46 | 65.52 | 72.99 | 87.95 | 24.18 | 56.07 | 90.04 | 41.71 | 65.87 | 95.52 | 44.45 | 69.99 | 65.85 | 63.16 | 65.51 |
| Freqfed | 77.71 | 55.17 | 66.44 | 86.63 | 25.37 | 56.50 | 77.39 | 61.84 | 69.61 | 78.42 | 40.47 | 59.44 | 75.39 | 61.84 | 68.61 |
| FedCPA | 77.43 | 78.76 | _78.09_ | 86.68 | 16.55 | 51.62 | 90.01 | 48.54 | 69.90 | 94.40 | 56.64 | 75.52 | 75.53 | 71.05 | 73.29 |
| G²uard | 70.49 | 66.00 | 68.25 | 85.54 | 83.73 | 84.64 | 88.89 | 65.13 | 77.01 | 93.06 | 58.98 | 76.02 | 73.12 | 73.56 | 73.34 |
| FedGTA | 81.03 | 55.12 | 68.08 | 87.40 | 47.88 | 67.64 | 89.55 | 40.91 | 65.23 | 95.02 | 53.05 | 74.04 | 78.88 | 71.38 | 75.13 |
| FGGP | 80.45 | 56.09 | 68.27 | 86.64 | 43.95 | 65.30 | 88.68 | 42.51 | 65.60 | 94.91 | 54.18 | 74.55 | 76.22 | 72.59 | 74.41 |
| FedTGE | 78.76 | 73.68 | 76.22 | 86.54 | 95.52 | _91.03_ | 89.23 | 77.37 | _83.30_ | 94.66 | 69.75 | _82.21_ | 75.50 | 80.26 | _77.89_ |
| FedTD | 80.03 | 76.66 | **78.35** | 87.42 | 94.98 | **91.20** | 90.55 | 78.20 | **84.38** | 95.08 | 70.07 | **82.58** | 77.49 | 85.92 | **81.71** |

Table 8: Comparison with baselines over three distinct scenarios on five mainstream datasets under Non-IID-Louvain setting with a malicious proportion of $\Upsilon = 0.5$ and a trigger type of Renyi.

| Scenarios | Citation Network | | | | | | Co-authorship | | | | | | Amazon-purchase | | |
| --- | --- | --- | --- | --- | --- | --- | --- | --- | --- | --- | --- | --- | --- | --- | --- |
| Datasets | Cora | | | Pubmed | | | CS | | | Physics | | | Photo | | |
| Metrics | $\mathcal{A}$ | $\mathcal{R}$ | $\mathcal{B}$ | $\mathcal{A}$ | $\mathcal{R}$ | $\mathcal{B}$ | $\mathcal{A}$ | $\mathcal{R}$ | $\mathcal{B}$ | $\mathcal{A}$ | $\mathcal{R}$ | $\mathcal{B}$ | $\mathcal{A}$ | $\mathcal{R}$ | $\mathcal{B}$ |
| FedAvg | 60.13 | 22.37 | 41.25 | 85.44 | 12.21 | 48.82 | 89.75 | 36.69 | 63.22 | 92.24 | 33.85 | 63.05 | 66.16 | 52.89 | 59.53 |
| Trimmed Median | 62.20 | 25.01 | 43.60 | 85.59 | 14.57 | 50.08 | 89.78 | 34.92 | 62.35 | 91.69 | 34.76 | 63.23 | 66.75 | 45.42 | 56.08 |
| Trimmed Mean | 62.26 | 29.26 | 45.78 | 85.89 | 14.46 | 50.18 | 89.46 | 33.69 | 61.57 | 93.55 | 57.33 | 75.44 | 66.62 | 49.24 | 57.93 |
| FoolsGold | 76.46 | 26.88 | 51.67 | 87.77 | 16.10 | 51.93 | 90.64 | 41.38 | 66.01 | 95.35 | 36.38 | 65.87 | 78.59 | 35.48 | 57.03 |
| DnC | 52.62 | 36.76 | 44.69 | 69.12 | 20.53 | 44.83 | 56.26 | 88.24 | 72.25 | 86.65 | 73.46 | 80.05 | 61.56 | 40.35 | 50.96 |
| SageFlow | 74.47 | 40.79 | 57.63 | 86.43 | 18.56 | 52.50 | 89.14 | 46.12 | 67.63 | 94.78 | 42.93 | 68.85 | 77.36 | 54.43 | 65.90 |
| MMA | 74.68 | 37.41 | 56.04 | 86.93 | 22.91 | 54.92 | 88.15 | 41.43 | 64.79 | 94.07 | 50.68 | 72.37 | 76.82 | 38.60 | 57.71 |
| RLR | 80.38 | 28.85 | 54.62 | 87.68 | 18.75 | 53.21 | 89.60 | 42.96 | 66.28 | 95.24 | 36.17 | 65.70 | 61.02 | 30.99 | 46.00 |
| Freqfed | 73.05 | 28.61 | 50.83 | 85.63 | 18.55 | 53.09 | 90.84 | 40.36 | 65.60 | 94.31 | 38.82 | 66.57 | 73.30 | 29.45 | 51.38 |
| FedCPA | 74.54 | 39.78 | 57.16 | 86.68 | 20.55 | 53.62 | 90.50 | 38.24 | 64.37 | 94.34 | 37.32 | 65.84 | 73.52 | 32.14 | 52.83 |
| G²uard | 63.46 | 44.63 | 54.05 | 84.58 | 34.73 | 59.66 | 88.02 | 68.51 | 78.27 | 92.30 | 63.01 | 77.66 | 70.23 | 76.69 | 73.46 |
| FedGTA | 70.02 | 29.87 | 49.95 | 85.09 | 22.03 | 53.56 | 89.92 | 39.98 | 64.95 | 89.01 | 38.72 | 63.87 | 72.88 | 51.09 | 61.99 |
| FGGP | 71.02 | 31.41 | 51.22 | 85.18 | 24.03 | 54.61 | 90.05 | 40.25 | 65.15 | 92.01 | 36.60 | 64.31 | 73.91 | 55.88 | 64.90 |
| FedTGE | 73.45 | 58.13 | 62.79 | 85.52 | 74.78 | 80.15 | 90.71 | 77.87 | 84.29 | 93.45 | 72.11 | 82.78 | 76.86 | 94.36 | 85.61 |
| FedTD | 74.50 | 59.09 | **66.80** | 87.09 | 75.23 | **81.16** | 90.91 | 77.99 | **84.45** | 94.09 | 73.09 | **83.59** | 77.01 | 94.55 | **85.78** |

Table 9: Comparison with baselines over three distinct scenarios on five mainstream datasets under IID setting with a malicious proportion of $\Upsilon = 0.3$ and a trigger type of GTA.

| Scenarios | Citation Network | | | | | | Co-authorship | | | | | | Amazon-purchase | | |
| --- | --- | --- | --- | --- | --- | --- | --- | --- | --- | --- | --- | --- | --- | --- | --- |
| Datasets | Cora | | | Pubmed | | | CS | | | Physics | | | Photo | | |
| Metrics | $\mathcal{A}$ | $\mathcal{R}$ | $\mathcal{B}$ | $\mathcal{A}$ | $\mathcal{R}$ | $\mathcal{B}$ | $\mathcal{A}$ | $\mathcal{R}$ | $\mathcal{B}$ | $\mathcal{A}$ | $\mathcal{R}$ | $\mathcal{B}$ | $\mathcal{A}$ | $\mathcal{R}$ | $\mathcal{B}$ |
| FedAvg | 76.54 | 25.96 | 51.25 | 84.86 | 9.70 | 47.28 | 81.92 | 52.82 | 67.37 | 91.42 | 22.86 | 57.14 | 70.32 | 48.32 | 59.32 |
| Trimmed Median | 77.49 | 31.06 | 54.26 | 84.63 | 12.58 | 48.60 | 82.16 | 57.94 | 70.05 | 91.55 | 25.39 | 58.47 | 71.76 | 57.91 | 64.83 |
| Trimmed Mean | 74.50 | 21.61 | 48.05 | 85.23 | 16.43 | 50.83 | 82.37 | 57.03 | 69.69 | 91.91 | 21.26 | 56.59 | 73.25 | 49.64 | 61.44 |
| FoolsGold | 77.56 | 23.60 | 50.58 | 87.15 | 14.07 | 50.61 | 88.78 | 46.48 | 67.63 | 92.68 | 38.56 | 65.62 | 70.97 | 41.39 | 56.18 |
| DnC | 65.90 | 54.82 | 60.36 | 73.92 | 30.83 | 52.37 | 56.25 | 88.42 | 72.33 | 77.51 | 56.64 | 67.07 | 60.79 | 93.08 | 76.93 |
| SageFlow | 77.95 | 27.49 | 52.72 | 87.04 | 14.70 | 50.87 | 88.09 | 68.80 | 78.45 | 92.54 | 52.69 | 72.62 | 71.49 | 40.51 | 55.99 |
| MMA | 77.72 | 44.95 | 61.34 | 86.11 | 21.91 | 54.01 | 88.43 | 73.92 | 81.17 | 92.63 | 37.62 | 65.12 | 69.49 | 56.13 | 62.81 |
| RLR | 81.65 | 25.48 | 53.56 | 86.88 | 15.20 | 51.04 | 89.28 | 73.42 | 81.35 | 92.99 | 39.56 | 66.27 | 70.83 | 48.45 | 59.64 |
| Freqfed | 79.91 | 27.62 | 53.76 | 85.86 | 13.60 | 49.73 | 88.49 | 53.92 | 71.21 | 92.95 | 38.96 | 65.95 | 71.20 | 46.89 | 59.05 |
| FedCPA | 76.79 | 22.00 | 49.40 | 86.42 | 12.62 | 49.52 | 88.64 | 58.24 | 73.44 | 92.55 | 38.16 | 65.35 | 70.97 | 36.36 | 53.67 |
| G²uard | 73.31 | 67.62 | 70.47 | 84.12 | 40.90 | 62.51 | 89.31 | 71.46 | 80.39 | 91.22 | 59.54 | 75.38 | 70.14 | 81.83 | 75.99 |
| FedGTA | 76.89 | 28.21 | 52.55 | 85.09 | 23.09 | 54.09 | 86.96 | 64.10 | 75.53 | 92.55 | 20.50 | 56.53 | 70.89 | 51.10 | 60.99 |
| FGGP | 77.92 | 30.81 | 54.37 | 86.04 | 11.91 | 48.98 | 85.09 | 52.73 | 68.91 | 91.89 | 28.05 | 59.97 | 71.99 | 58.06 | 65.03 |
| FedTGE | 78.23 | 73.23 | 75.73 | 85.56 | 53.34 | 69.45 | 88.53 | 85.82 | 87.18 | 92.39 | 70.37 | 81.38 | 72.46 | 95.56 | 84.01 |
| FedTD | 80.71 | 76.08 | **78.40** | 86.91 | 53.08 | **70.00** | 89.02 | 85.89 | **87.46** | 93.20 | 71.03 | **82.12** | 72.67 | 95.49 | **84.08** |

Table 10: Comparison with baselines over three distinct scenarios on five mainstream datasets under IID setting with a malicious proportion of $\Upsilon = 0.3$ and a trigger type of WS.

| Scenarios | Citation Network | | | | | | Co-authorship | | | | | | Amazon-purchase | | |
| --- | --- | --- | --- | --- | --- | --- | --- | --- | --- | --- | --- | --- | --- | --- | --- |
| Datasets | Cora | | | Pubmed | | | CS | | | Physics | | | Photo | | |
| Metrics | $\mathcal{A}$ | $\mathcal{R}$ | $\mathcal{B}$ | $\mathcal{A}$ | $\mathcal{R}$ | $\mathcal{B}$ | $\mathcal{A}$ | $\mathcal{R}$ | $\mathcal{B}$ | $\mathcal{A}$ | $\mathcal{R}$ | $\mathcal{B}$ | $\mathcal{A}$ | $\mathcal{R}$ | $\mathcal{B}$ |
| FedAvg | 74.81 | 72.24 | 73.53 | 82.84 | 79.47 | 81.16 | 83.56 | 74.13 | 78.85 | 91.32 | 60.53 | 75.93 | 71.53 | 76.51 | 74.02 |
| Trimmed Median | 74.81 | 72.24 | 73.53 | 82.63 | 68.89 | 75.76 | 87.72 | 75.29 | 81.51 | 90.45 | 55.91 | 73.18 | 78.32 | 75.25 | 76.79 |
| Trimmed Mean | 78.84 | 89.88 | 84.36 | 83.12 | 69.61 | 76.37 | 87.83 | 74.33 | 81.08 | 91.07 | 56.24 | 73.66 | 79.26 | 76.15 | 77.71 |
| FoolsGold | 80.41 | 82.22 | 81.32 | 84.52 | 75.78 | 80.15 | 88.41 | 75.62 | 82.02 | 91.59 | 65.79 | 78.69 | 77.57 | 86.18 | 81.88 |
| DnC | 62.39 | 83.90 | 73.15 | 76.25 | 69.68 | 72.97 | 75.25 | 93.94 | 84.60 | 75.84 | 75.93 | 75.89 | 64.28 | 96.28 | 80.28 |
| SageFlow | 79.31 | 86.90 | 83.11 | 83.60 | 83.84 | 83.72 | 87.04 | 74.28 | 80.66 | 91.50 | 64.74 | 78.12 | 75.26 | 70.76 | 73.01 |
| MMA | 78.60 | 88.80 | 83.70 | 84.32 | 77.53 | 80.93 | 84.27 | 76.95 | 80.61 | 91.88 | 62.80 | 77.34 | 78.42 | 72.62 | 75.52 |
| RLR | 79.32 | 94.72 | 87.02 | 84.84 | 74.30 | 79.57 | 88.12 | 74.50 | 81.31 | 92.14 | 64.04 | 78.09 | 77.69 | 73.88 | 75.79 |
| Freqfed | 80.91 | 70.28 | 75.60 | 84.25 | 73.82 | 79.04 | 88.41 | 87.15 | 87.78 | 90.81 | 66.31 | 78.56 | 77.96 | 75.99 | 76.98 |
| FedCPA | 79.70 | 88.00 | 83.85 | 83.31 | 76.59 | 79.95 | 87.74 | 77.15 | 82.45 | 92.07 | 67.33 | 79.70 | 78.52 | 74.34 | 76.43 |
| G²uard | 75.49 | 92.36 | 83.93 | 82.82 | 76.24 | 79.53 | 87.54 | 76.79 | 82.17 | 90.93 | 66.68 | 78.81 | 74.72 | 72.51 | 73.62 |
| FedGTA | 77.59 | 80.41 | 79.00 | 83.33 | 77.60 | 80.47 | 86.69 | 76.63 | 81.66 | 91.59 | 63.81 | 77.70 | 73.09 | 77.72 | 75.41 |
| FGGP | 77.51 | 81.91 | 79.71 | 82.68 | 77.49 | 80.09 | 87.41 | 77.52 | 82.47 | 90.62 | 63.31 | 76.97 | 76.72 | 75.69 | 76.21 |
| FedTGE | 79.26 | 89.48 | 84.37 | 83.73 | 87.69 | 85.71 | 88.33 | 90.46 | 89.40 | 91.95 | 83.19 | 87.57 | 76.19 | 94.62 | 85.41 |
| FedTD | 79.58 | 89.72 | **84.65** | 84.50 | 87.83 | **86.17** | 88.72 | 90.41 | **89.57** | 92.39 | 84.58 | **88.49** | 78.72 | 94.51 | **86.62** |

Table 11: Comparison with baselines over three distinct scenarios on five mainstream datasets under IID setting with a malicious proportion of $\Upsilon = 0.3$ and a trigger type of BA.

| Scenarios | Citation Network | | | | | | Co-authorship | | | | | | Amazon-purchase | | |
|---|---|---|---|---|---|---|---|---|---|---|---|---|---|---|---|
| Datasets | Cora | | | Pubmed | | | CS | | | Physics | | | Photo | | |
| Metrics | $\mathcal{A}$ | $\mathcal{R}$ | $\mathcal{B}$ | $\mathcal{A}$ | $\mathcal{R}$ | $\mathcal{B}$ | $\mathcal{A}$ | $\mathcal{R}$ | $\mathcal{B}$ | $\mathcal{A}$ | $\mathcal{R}$ | $\mathcal{B}$ | $\mathcal{A}$ | $\mathcal{R}$ | $\mathcal{B}$ |
| FedAvg | 76.55 | 77.03 | 76.79 | 83.05 | 83.20 | 83.13 | 88.51 | 69.74 | 79.13 | 90.45 | 61.41 | 75.93 | 76.18 | 74.77 | 75.48 |
| Trimmed Median | 77.68 | 86.05 | 81.86 | 82.71 | 67.13 | 74.92 | 85.91 | 75.60 | 80.76 | 90.36 | 55.76 | 73.06 | 77.77 | 76.15 | 76.96 |
| Trimmed Mean | 76.17 | 80.20 | 78.18 | 83.85 | 73.13 | 78.49 | 87.71 | 74.39 | 81.05 | 91.08 | 59.53 | 75.31 | 78.49 | 76.60 | 77.55 |
| FoolsGold | 79.62 | 85.66 | 82.64 | 83.71 | 69.37 | 76.54 | 87.25 | 76.95 | 82.10 | 91.88 | 64.77 | 78.33 | 78.67 | 74.85 | 76.76 |
| DnC | 70.94 | 90.83 | 80.89 | 80.72 | 70.45 | 75.59 | 71.23 | 75.39 | 73.31 | 74.56 | 68.65 | 71.61 | 68.42 | 76.15 | 72.29 |
| SageFlow | 80.16 | 88.92 | 84.54 | 85.35 | 72.89 | 79.12 | 87.55 | 76.89 | 82.22 | 89.75 | 65.65 | 77.70 | 79.74 | 76.60 | 78.17 |
| MMA | 78.47 | 90.03 | 84.25 | 83.17 | 75.77 | 79.47 | 85.47 | 75.31 | 80.39 | 89.84 | 58.00 | 73.92 | 76.17 | 74.38 | 75.28 |
| RLR | 77.29 | 92.73 | 85.01 | 85.05 | 80.27 | 82.66 | 88.67 | 77.15 | 82.91 | 92.04 | 61.79 | 76.92 | 78.05 | 75.70 | 76.88 |
| Freqfed | 78.94 | 90.25 | 84.60 | 83.37 | 77.71 | 80.54 | 88.19 | 74.09 | 81.14 | 91.94 | 58.07 | 75.01 | 78.94 | 75.73 | 77.34 |
| FedCPA | 77.56 | 84.93 | 81.25 | 83.56 | 76.90 | 80.23 | 87.31 | 71.20 | 79.26 | 91.97 | 58.42 | 75.20 | 79.47 | 74.83 | 77.15 |
| $G^2$uard | 77.46 | 85.65 | 81.55 | 82.34 | 84.16 | 83.25 | 87.85 | 72.40 | 80.13 | 91.71 | 64.17 | 77.94 | 75.41 | 74.83 | 75.12 |
| FedGTA | 79.05 | 76.03 | 77.54 | 84.59 | 75.17 | 79.88 | 88.30 | 71.35 | 79.83 | 84.06 | 58.27 | 71.17 | 78.50 | 70.75 | 74.63 |
| FGGP | 78.34 | 80.01 | 79.18 | 82.49 | 77.13 | 79.81 | 88.05 | 72.11 | 80.08 | 88.42 | 58.61 | 73.52 | 77.69 | 71.72 | 74.71 |
| FedTGE | 79.48 | 90.08 | 84.78 | 84.77 | 89.06 | 86.92 | 88.63 | 84.39 | 86.51 | 91.05 | 73.17 | 82.11 | 78.28 | 86.21 | 82.25 |
| FedTD | 79.83 | 92.78 | **86.31** | 85.39 | 90.63 | **88.01** | 88.52 | 86.09 | **87.31** | 91.22 | 73.54 | **82.38** | 78.95 | 85.79 | **82.37** |

Table 12: Comparison with baselines over three distinct scenarios on five mainstream datasets under IID setting with a malicious proportion of $\Upsilon = 0.3$ and a trigger type of Opt-GDBA.

| Scenarios | Citation Network | | | | | | Co-authorship | | | | | | Amazon-purchase | | |
|---|---|---|---|---|---|---|---|---|---|---|---|---|---|---|---|
| Datasets | Cora | | | Pubmed | | | CS | | | Physics | | | Photo | | |
| Metrics | $\mathcal{A}$ | $\mathcal{R}$ | $\mathcal{B}$ | $\mathcal{A}$ | $\mathcal{R}$ | $\mathcal{B}$ | $\mathcal{A}$ | $\mathcal{R}$ | $\mathcal{B}$ | $\mathcal{A}$ | $\mathcal{R}$ | $\mathcal{B}$ | $\mathcal{A}$ | $\mathcal{R}$ | $\mathcal{B}$ |
| FedAvg | 59.68 | 58.56 | 59.12 | 67.94 | 65.43 | 66.69 | 69.17 | 58.98 | 64.08 | 75.12 | 43.86 | 59.49 | 56.34 | 64.62 | 60.48 |
| Trimmed Median | 58.97 | 61.18 | 60.08 | 67.31 | 53.34 | 60.33 | 70.58 | 59.64 | 65.11 | 72.52 | 39.91 | 56.22 | 64.39 | 60.12 | 62.26 |
| Trimmed Mean | 65.73 | 72.91 | 69.32 | 66.98 | 55.31 | 61.15 | 69.08 | 57.19 | 63.14 | 74.92 | 40.99 | 57.96 | 63.65 | 62.93 | 63.29 |
| FoolsGold | 65.75 | 66.59 | 66.17 | 68.23 | 60.13 | 64.18 | 69.34 | 60.70 | 65.02 | 75.67 | 48.35 | 62.01 | 62.15 | 70.18 | 66.17 |
| DnC | 50.22 | 68.65 | 59.44 | 60.72 | 54.48 | 57.60 | 58.63 | 75.89 | 67.26 | 60.74 | 59.64 | 60.19 | 48.96 | 77.23 | 63.10 |
| SageFlow | 65.99 | 72.49 | 69.24 | 67.88 | 67.74 | 67.81 | 69.82 | 58.33 | 64.08 | 74.95 | 50.86 | 62.91 | 60.65 | 56.95 | 58.80 |
| MMA | 63.91 | 71.33 | 67.62 | 67.98 | 61.72 | 64.85 | 67.74 | 60.17 | 63.95 | 75.14 | 49.08 | 62.11 | 62.88 | 58.16 | 60.52 |
| RLR | 64.05 | 75.93 | 69.99 | 68.51 | 59.67 | 64.09 | 69.68 | 58.79 | 64.24 | 75.32 | 50.19 | 62.75 | 62.15 | 59.16 | 60.66 |
| Freqfed | 66.94 | 55.76 | 61.35 | 68.80 | 58.10 | 63.45 | 70.01 | 70.32 | 70.17 | 74.99 | 50.56 | 62.78 | 63.74 | 60.61 | 62.18 |
| FedCPA | 64.81 | 70.44 | 67.63 | 67.02 | 61.72 | 64.37 | 69.18 | 60.75 | 64.97 | 75.01 | 52.40 | 63.71 | 63.72 | 59.73 | 61.73 |
| $G^2$uard | 60.61 | 74.60 | 67.61 | 66.94 | 60.67 | 63.80 | 69.26 | 60.92 | 65.09 | 72.99 | 51.49 | 62.24 | 59.97 | 58.61 | 59.29 |
| FedGTA | 62.84 | 64.47 | 63.66 | 67.57 | 61.63 | 64.60 | 69.03 | 60.83 | 64.96 | 74.34 | 48.46 | 61.40 | 57.48 | 62.86 | 60.17 |
| FGGP | 62.45 | 65.69 | 64.07 | 66.76 | 61.80 | 64.28 | 69.51 | 60.49 | 65.00 | 72.59 | 48.24 | 60.42 | 61.83 | 60.78 | 61.31 |
| FedTGE | 65.31 | 72.43 | 68.87 | 67.10 | 69.65 | 68.38 | 70.03 | 71.91 | 70.97 | 75.64 | 66.27 | 70.96 | 60.92 | 75.82 | 68.37 |
| FedTD | 71.92 | 80.25 | **76.09** | 76.52 | 74.51 | **75.52** | 78.28 | 79.03 | **78.66** | 83.91 | 73.85 | **78.88** | 71.55 | 85.01 | **78.28** |

Table 13: Comparison with baselines over three distinct scenarios on five mainstream datasets under Non-IID-Louvain setting with a malicious proportion of $\Upsilon = 0.3$ and a trigger type of GTA.

| Scenarios | Citation Network | | | | | | Co-authorship | | | | | | Amazon-purchase | | |
|---|---|---|---|---|---|---|---|---|---|---|---|---|---|---|---|
| Datasets | Cora | | | Pubmed | | | CS | | | Physics | | | Photo | | |
| Metrics | $\mathcal{A}$ | $\mathcal{R}$ | $\mathcal{B}$ | $\mathcal{A}$ | $\mathcal{R}$ | $\mathcal{B}$ | $\mathcal{A}$ | $\mathcal{R}$ | $\mathcal{B}$ | $\mathcal{A}$ | $\mathcal{R}$ | $\mathcal{B}$ | $\mathcal{A}$ | $\mathcal{R}$ | $\mathcal{B}$ |
| FedAvg | 78.54 | 27.96 | 53.25 | 86.86 | 11.70 | 49.28 | 83.92 | 54.82 | 69.37 | 93.42 | 24.86 | 59.14 | 72.32 | 50.32 | 61.32 |
| Trimmed Median | 79.49 | 33.06 | 56.26 | 86.63 | 14.58 | 50.60 | 84.16 | 59.94 | 72.05 | 93.55 | 27.39 | 60.47 | 73.76 | 59.91 | 66.83 |
| Trimmed Mean | 76.50 | 23.61 | 50.05 | 87.23 | 18.43 | 52.83 | 84.37 | 59.03 | 71.69 | 93.91 | 23.26 | 58.59 | 75.25 | 51.64 | 63.44 |
| FoolsGold | 79.56 | 25.60 | 52.58 | 89.15 | 16.07 | 52.61 | 90.78 | 48.48 | 69.63 | 94.68 | 40.56 | 67.62 | 72.97 | 43.39 | 58.18 |
| DnC | 67.90 | 56.82 | 62.36 | 75.92 | 32.83 | 54.37 | 58.25 | 90.42 | 74.33 | 79.51 | 58.64 | 69.07 | 62.79 | 95.08 | 78.93 |
| SageFlow | 79.95 | 29.49 | 54.72 | 89.04 | 16.70 | 52.87 | 90.09 | 70.80 | 80.45 | 94.54 | 54.69 | 74.62 | 73.49 | 42.51 | 57.99 |
| MMA | 79.72 | 46.95 | 63.34 | 88.11 | 23.91 | 56.01 | 90.43 | 75.92 | 83.17 | 94.63 | 39.62 | 67.12 | 71.49 | 58.13 | 64.81 |
| RLR | 83.65 | 27.48 | 55.56 | 88.88 | 17.20 | 53.04 | 91.28 | 75.42 | 83.35 | 94.99 | 41.56 | 68.27 | 72.83 | 50.45 | 61.64 |
| Freqfed | 81.91 | 29.62 | 55.76 | 87.86 | 15.60 | 51.73 | 90.49 | 55.92 | 73.21 | 94.95 | 40.96 | 67.95 | 73.20 | 48.89 | 61.05 |
| FedCPA | 78.79 | 24.00 | 51.40 | 88.42 | 14.62 | 51.52 | 90.64 | 60.24 | 75.44 | 94.55 | 40.16 | 67.35 | 72.97 | 38.36 | 55.67 |
| $G^2$uard | 75.31 | 69.62 | 72.47 | 86.12 | 42.90 | 64.51 | 91.31 | 73.46 | 82.39 | 93.22 | 61.54 | 77.38 | 72.14 | 83.83 | 77.99 |
| FedGTA | 78.89 | 30.21 | 54.55 | 87.09 | 25.09 | 56.09 | 88.96 | 66.10 | 77.53 | 94.55 | 22.50 | 58.53 | 72.89 | 53.08 | 62.99 |
| FGGP | 79.92 | 32.81 | 56.37 | 88.04 | 13.91 | 50.98 | 87.09 | 54.73 | 70.91 | 93.89 | 30.05 | 61.97 | 73.99 | 60.06 | 67.03 |
| FedTGE | 80.23 | 75.23 | 77.73 | 87.56 | 55.34 | 71.45 | 90.53 | 87.82 | 89.18 | 94.39 | 72.37 | 83.38 | 74.46 | 97.56 | 86.01 |
| FedTD | 82.71 | 78.08 | **80.40** | 88.91 | 55.08 | **72.00** | 91.02 | 87.89 | **89.46** | 95.20 | 73.03 | **84.12** | 74.67 | 97.49 | **86.08** |

Table 14: Comparison with baselines over three distinct scenarios on five mainstream datasets under Non-IID-Louvain setting with a malicious proportion of $\Upsilon = 0.3$ and a trigger type of WS.

| Scenarios | Citation Network | | | | | | Co-authorship | | | | | | Amazon-purchase | | |
|---|---|---|---|---|---|---|---|---|---|---|---|---|---|---|---|
| Datasets | Cora | | | Pubmed | | | CS | | | Physics | | | Photo | | |
| Metrics | $\mathcal{A}$ | $\mathcal{R}$ | $\mathcal{B}$ | $\mathcal{A}$ | $\mathcal{R}$ | $\mathcal{B}$ | $\mathcal{A}$ | $\mathcal{R}$ | $\mathcal{B}$ | $\mathcal{A}$ | $\mathcal{R}$ | $\mathcal{B}$ | $\mathcal{A}$ | $\mathcal{R}$ | $\mathcal{B}$ |
| FedAvg | 77.31 | 74.74 | 76.03 | 85.34 | 81.97 | 83.66 | 86.06 | 76.63 | 81.35 | 93.82 | 63.03 | 78.42 | 74.03 | 79.01 | 76.52 |
| Trimmed Median | 77.31 | 74.74 | 76.03 | 85.13 | 71.39 | 78.26 | 90.22 | 77.79 | 84.01 | 92.95 | 58.41 | 75.68 | 80.82 | 77.75 | 79.29 |
| Trimmed Mean | 81.34 | 92.38 | 86.86 | 85.62 | 72.11 | 78.87 | 90.33 | 76.83 | 83.58 | 93.57 | 58.74 | 76.16 | 81.76 | 78.65 | 80.21 |
| FoolsGold | 82.91 | 84.72 | 83.82 | 87.02 | 78.28 | 82.65 | 90.91 | 78.12 | 84.52 | 94.09 | 68.29 | 81.19 | 80.07 | 88.68 | 84.38 |
| DnC | 64.89 | 86.40 | 75.65 | 78.75 | 72.18 | 75.46 | 77.75 | 96.44 | 87.10 | 78.34 | 78.43 | 78.38 | 66.78 | 98.78 | 82.78 |
| SageFlow | 81.81 | 89.40 | 85.61 | 86.10 | 86.34 | 86.22 | 89.54 | 76.78 | 83.16 | 94.00 | 67.24 | 80.62 | 77.76 | 73.26 | 75.51 |
| MMA | 81.10 | 91.30 | 86.20 | 86.82 | 80.03 | 83.42 | 86.77 | 79.45 | 83.11 | 94.38 | 65.30 | 79.84 | 80.92 | 75.12 | 78.00 |
| RLR | 81.82 | 97.22 | 89.52 | 87.34 | 76.80 | 82.07 | 90.62 | 77.00 | 83.81 | 94.64 | 66.54 | 80.59 | 80.19 | 76.38 | 78.29 |
| Freqfed | 83.41 | 72.78 | 78.10 | 86.75 | 76.32 | 81.53 | 90.91 | 89.65 | 90.28 | 93.31 | 68.81 | 81.06 | 80.46 | 78.49 | 79.48 |
| FedCPA | 82.20 | 90.50 | 86.35 | 85.81 | 79.09 | 82.45 | 90.24 | 79.65 | 84.95 | 94.57 | 69.83 | 82.20 | 81.02 | 76.84 | 78.93 |
| G²uard | 77.99 | 94.86 | 86.42 | 85.32 | 78.74 | 82.03 | 90.04 | 79.29 | 84.67 | 93.43 | 69.18 | 81.31 | 77.22 | 75.01 | 76.12 |
| FedGTA | 80.09 | 82.91 | 81.50 | 85.83 | 80.10 | 82.97 | 89.19 | 79.13 | 84.16 | 94.09 | 66.31 | 80.20 | 75.59 | 80.22 | 77.91 |
| FGGP | 80.01 | 84.41 | 82.21 | 85.18 | 79.99 | 82.59 | 89.91 | 80.02 | 84.97 | 93.12 | 65.81 | 79.47 | 79.22 | 78.19 | 78.71 |
| FedTGE | 81.76 | 91.98 | 86.87 | 86.23 | 90.19 | 88.21 | 90.83 | 92.96 | 91.90 | 94.45 | 85.69 | 90.07 | 78.69 | 97.12 | 87.91 |
| FedTD | 82.08 | 92.22 | **87.15** | 87.00 | 90.33 | **88.67** | 91.22 | 92.91 | **92.07** | 94.89 | 87.08 | **90.99** | 81.22 | 97.01 | **89.12** |

Table 15: Comparison with baselines over three distinct scenarios on five mainstream datasets under Non-IID-Louvain setting with a malicious proportion of $\Upsilon = 0.3$ and a trigger type of BA.

| Scenarios | Citation Network | | | | | | Co-authorship | | | | | | Amazon-purchase | | |
|---|---|---|---|---|---|---|---|---|---|---|---|---|---|---|---|
| Datasets | Cora | | | Pubmed | | | CS | | | Physics | | | Photo | | |
| Metrics | $\mathcal{A}$ | $\mathcal{R}$ | $\mathcal{B}$ | $\mathcal{A}$ | $\mathcal{R}$ | $\mathcal{B}$ | $\mathcal{A}$ | $\mathcal{R}$ | $\mathcal{B}$ | $\mathcal{A}$ | $\mathcal{R}$ | $\mathcal{B}$ | $\mathcal{A}$ | $\mathcal{R}$ | $\mathcal{B}$ |
| FedAvg | 79.05 | 79.53 | 79.29 | 85.55 | 85.70 | 85.63 | 91.01 | 72.24 | 81.63 | 92.95 | 63.91 | 78.43 | 78.68 | 77.27 | 77.98 |
| Trimmed Median | 80.18 | 88.55 | 84.36 | 85.21 | 69.63 | 77.42 | 88.41 | 78.10 | 83.26 | 92.86 | 58.26 | 75.56 | 80.27 | 78.65 | 79.46 |
| Trimmed Mean | 78.67 | 82.70 | 80.68 | 86.35 | 75.63 | 80.99 | 90.21 | 76.89 | 83.55 | 93.58 | 62.03 | 77.81 | 80.99 | 79.10 | 80.05 |
| FoolsGold | 82.12 | 88.16 | 85.14 | 86.21 | 71.87 | 79.04 | 89.75 | 79.45 | 84.61 | 94.38 | 67.27 | 80.83 | 81.17 | 77.35 | 79.26 |
| DnC | 73.44 | 93.33 | 83.39 | 83.22 | 72.95 | 78.09 | 73.73 | 77.89 | 75.81 | 77.06 | 71.15 | 74.11 | 70.92 | 78.65 | 74.79 |
| SageFlow | 82.66 | 91.42 | 87.04 | 87.85 | 75.39 | 81.62 | 90.05 | 79.39 | 84.72 | 92.25 | 68.15 | 80.20 | 82.24 | 79.10 | 80.67 |
| MMA | 80.97 | 92.53 | 86.75 | 85.67 | 78.27 | 81.96 | 87.97 | 77.81 | 82.89 | 92.34 | 60.50 | 76.42 | 78.67 | 76.88 | 77.78 |
| RLR | 79.79 | 95.23 | 87.51 | 87.55 | 82.77 | 85.16 | 91.17 | 79.65 | 85.41 | 94.54 | 64.29 | 79.42 | 80.55 | 78.20 | 79.38 |
| Freqfed | 81.44 | 92.75 | 87.10 | 85.87 | 80.21 | 83.04 | 90.69 | 76.59 | 83.64 | 94.44 | 60.57 | 77.50 | 81.44 | 78.23 | 79.84 |
| FedCPA | 80.06 | 87.43 | 83.75 | 86.06 | 79.40 | 82.73 | 89.81 | 73.70 | 81.76 | 94.47 | 60.92 | 77.70 | 81.97 | 77.33 | 79.65 |
| G²uard | 79.96 | 88.15 | 84.05 | 84.84 | 86.66 | 85.75 | 90.35 | 74.90 | 82.63 | 94.21 | 66.67 | 80.44 | 77.91 | 77.33 | 77.62 |
| FedGTA | 81.55 | 78.53 | 80.04 | 87.09 | 77.67 | 82.38 | 90.80 | 73.85 | 82.33 | 86.56 | 60.77 | 73.67 | 81.00 | 73.25 | 77.13 |
| FGGP | 80.84 | 82.51 | 81.68 | 84.99 | 79.63 | 82.31 | 90.55 | 74.61 | 82.58 | 90.92 | 61.11 | 76.02 | 80.19 | 74.22 | 77.21 |
| FedTGE | 81.98 | 92.58 | 87.28 | 87.27 | 91.56 | 89.42 | 91.13 | 86.89 | 89.01 | 93.55 | 75.67 | 84.61 | 80.78 | 88.71 | 84.75 |
| FedTD | 82.33 | 95.28 | **88.81** | 87.89 | 93.13 | **90.51** | 91.02 | 88.59 | **89.81** | 93.72 | 76.04 | **84.88** | 81.45 | 88.29 | **84.87** |

Table 16: Comparison with baselines over three distinct scenarios on five mainstream datasets under Non-IID-Louvain setting with a malicious proportion of $\Upsilon = 0.3$ and a trigger type of Opt-GDBA.

| Scenarios | Citation Network | | | | | | Co-authorship | | | | | | Amazon-purchase | | |
|---|---|---|---|---|---|---|---|---|---|---|---|---|---|---|---|
| Datasets | Cora | | | Pubmed | | | CS | | | Physics | | | Photo | | |
| Metrics | $\mathcal{A}$ | $\mathcal{R}$ | $\mathcal{B}$ | $\mathcal{A}$ | $\mathcal{R}$ | $\mathcal{B}$ | $\mathcal{A}$ | $\mathcal{R}$ | $\mathcal{B}$ | $\mathcal{A}$ | $\mathcal{R}$ | $\mathcal{B}$ | $\mathcal{A}$ | $\mathcal{R}$ | $\mathcal{B}$ |
| FedAvg | 60.02 | 58.87 | 59.45 | 68.46 | 65.86 | 67.16 | 70.91 | 60.88 | 65.90 | 75.42 | 48.92 | 62.17 | 57.81 | 62.01 | 59.91 |
| Trimmed Median | 60.94 | 56.92 | 58.93 | 68.73 | 54.00 | 61.37 | 72.19 | 61.94 | 67.07 | 75.13 | 44.42 | 59.78 | 63.78 | 62.77 | 63.28 |
| Trimmed Mean | 65.86 | 73.16 | 69.51 | 68.64 | 55.80 | 62.22 | 72.70 | 60.84 | 66.77 | 75.64 | 44.96 | 60.30 | 64.97 | 62.98 | 63.97 |
| FoolsGold | 65.45 | 67.74 | 66.60 | 69.61 | 61.08 | 65.34 | 72.85 | 61.70 | 67.28 | 75.39 | 52.63 | 64.01 | 62.07 | 71.03 | 66.55 |
| DnC | 49.90 | 67.77 | 58.84 | 61.66 | 55.72 | 58.69 | 60.11 | 77.73 | 68.92 | 60.62 | 62.51 | 61.57 | 50.77 | 78.98 | 64.88 |
| SageFlow | 65.93 | 70.07 | 68.00 | 69.61 | 67.87 | 68.74 | 71.88 | 60.74 | 66.31 | 75.66 | 51.98 | 63.82 | 61.23 | 57.75 | 59.49 |
| MMA | 65.79 | 72.63 | 69.21 | 69.60 | 63.68 | 66.64 | 69.36 | 62.71 | 66.04 | 75.54 | 48.40 | 61.97 | 63.34 | 58.38 | 60.86 |
| RLR | 65.81 | 77.81 | 71.81 | 70.20 | 59.30 | 64.75 | 72.20 | 60.59 | 66.40 | 75.13 | 50.15 | 62.64 | 62.04 | 59.01 | 60.53 |
| Freqfed | 66.86 | 56.84 | 61.85 | 69.50 | 59.00 | 64.25 | 73.49 | 71.55 | 72.52 | 74.44 | 53.73 | 64.09 | 63.20 | 62.94 | 63.07 |
| FedCPA | 65.76 | 71.77 | 68.77 | 68.17 | 62.14 | 65.16 | 72.01 | 62.91 | 67.46 | 75.64 | 53.74 | 64.69 | 64.62 | 60.31 | 62.47 |
| G²uard | 62.87 | 75.31 | 69.09 | 68.62 | 61.13 | 64.88 | 71.95 | 62.68 | 67.31 | 74.31 | 53.71 | 64.01 | 60.93 | 59.69 | 60.31 |
| FedGTA | 64.85 | 65.70 | 65.28 | 69.23 | 63.85 | 66.54 | 71.95 | 63.06 | 67.51 | 75.19 | 49.24 | 62.22 | 58.88 | 62.65 | 60.77 |
| FGGP | 64.84 | 66.35 | 65.60 | 67.06 | 63.11 | 65.09 | 71.40 | 62.85 | 67.12 | 74.50 | 49.05 | 61.78 | 62.37 | 61.90 | 62.14 |
| FedTGE | 65.91 | 73.16 | 69.54 | 70.55 | 73.57 | 72.06 | 72.95 | 74.33 | 73.64 | 75.80 | 68.18 | 71.99 | 60.81 | 77.91 | 69.36 |
| FedTD | 73.19 | 84.61 | **78.90** | 79.23 | 81.09 | **80.16** | 80.28 | 81.85 | **81.07** | 82.09 | 75.66 | **78.88** | 70.72 | 84.44 | **77.58** |

Table 17: Comparison with different GNN backbones with a malicious proportion of $\Upsilon = 0.3$ and a trigger type of Renyi.

| Scenarios | Citation Network | | | | | | Co-authorship | | | | | | Amazon-purchase | | |
|---|---|---|---|---|---|---|---|---|---|---|---|---|---|---|---|
| Datasets | Cora | | | Pubmed | | | CS | | | Physics | | | Photo | | |
| Metrics | $\mathcal{A}$ | $\mathcal{R}$ | $\mathcal{B}$ | $\mathcal{A}$ | $\mathcal{R}$ | $\mathcal{B}$ | $\mathcal{A}$ | $\mathcal{R}$ | $\mathcal{B}$ | $\mathcal{A}$ | $\mathcal{R}$ | $\mathcal{B}$ | $\mathcal{A}$ | $\mathcal{R}$ | $\mathcal{B}$ |
| | IID Setting | | | | | | | | | | | | | | |
| FedTD (GCN) | 76.37 | 72.91 | 74.64 | 85.09 | 60.31 | 72.70 | 85.69 | 71.20 | 78.45 | 93.80 | 62.65 | 78.23 | 82.50 | 97.04 | 89.77 |
| FedTD (GAT) | 77.01 | 70.75 | 73.88 | 85.52 | 60.07 | 72.80 | 85.90 | 71.31 | 78.61 | 93.55 | 62.81 | 78.18 | 82.58 | 96.74 | 89.66 |
| FedTD (GraphSage) | 75.37 | 71.55 | 73.46 | 84.62 | 60.00 | 72.31 | 85.23 | 70.81 | 78.02 | 93.70 | 62.29 | 78.00 | 82.56 | 96.89 | 89.73 |
| | Non-IID-Louvain Setting | | | | | | | | | | | | | | |
| FedTD (GCN) | 78.78 | 58.02 | 68.40 | 87.21 | 69.07 | 78.14 | 89.92 | 73.89 | 81.91 | 95.09 | 61.09 | 78.09 | 77.91 | 94.69 | 86.30 |
| FedTD (GAT) | 78.09 | 58.15 | 68.12 | 87.06 | 69.13 | 78.10 | 89.12 | 73.34 | 79.18 | 95.26 | 61.03 | 78.15 | 77.59 | 94.33 | 85.96 |
| FedTD (GraphSage) | 78.41 | 59.10 | 68.76 | 87.30 | 69.41 | 78.22 | 88.70 | 73.22 | 80.96 | 93.97 | 63.12 | 78.55 | 77.84 | 93.95 | 85.90 |

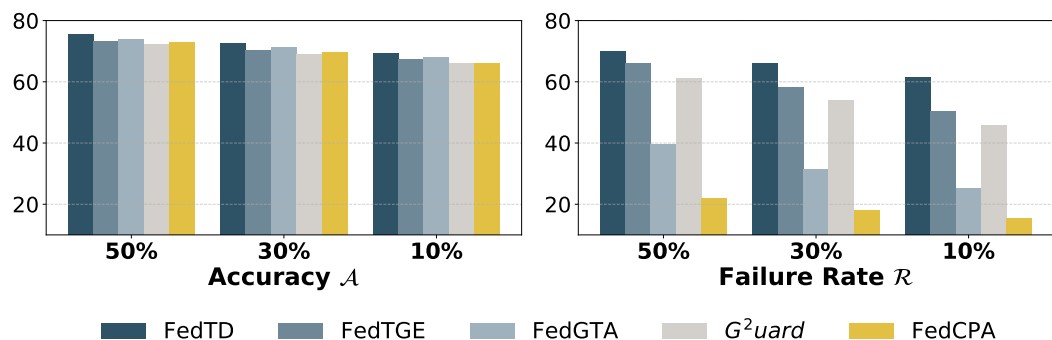

Figure 7: Client selection analysis on the Cora dataset under IID setting with a malicious proportion of $\Upsilon = 0.3$ and a trigger type of Renyi.

FedTD delivers satisfactory performance across various GNN architectures, regardless of whether the settings are IID or Non-IID-Louvain.

### A.9 THREAT MODEL

We adopt threat model following previous studies (Wan et al., 2025; Xu et al., 2021; Liu et al., 2023; Wan et al., 2025). Assume there are $K$ clients in total, among which $M$ ($M \leq K$) are malicious. Each malicious client independently executes a backdoor attack on its own model. The primary objective of a backdoor attack is to manipulate the model so that it misclassifies specific pre-defined labels (referred to as target labels) within the poisoned data samples, while maintaining high accuracy on clean data. **Attack Knowledge:** In this context, it is assumed that each malicious client has complete knowledge of its own training data and can generate triggers accordingly. This assumption is practical, given that clients have full control over their local data. **Attacker Capability:** A malicious client can inject triggers into its training datasets, subject to predefined constraints such as trigger size and the poisoning rate. The purpose of this is to contaminate the training data. However, the attacker cannot manipulate the server-side aggregation process or interfere with the training processes or models of other clients.

Mathematically, the formal attack objective for each malicious client $c_i$ during round $t$ can be defined as follows:

$$w_k^{*t} =$$

$$\arg\min_{w_k^t} \frac{1}{|V_k|} \left[ \sum_{v_i \in V_k^p} \mathcal{L}\left(f_\theta\left(x_i, g_\tau \circ N\left(v_i\right); w_k^{t-1}\right), \tau\right) + \sum_{v_i \in V_k^c} \mathcal{L}\left(f_\theta\left(x_i, N\left(v_i\right); w_k^{t-1}\right), y_i\right) \right],$$

$$\forall v_i \in V_k^p, \quad N_\tau = |g_\tau| \leq \triangle_g \quad \text{and} \quad \rho = \frac{|V_k^p|}{|V_k|} \leq \triangle_p,$$

(21)

where $V_k^p$ refers to the set of poisoned nodes and $V_k^c$ corresponds to the clean node set for client $c_k$. The GNN model is represented as $f_\theta(x_i, N(v_i); w_k)$ where $N(v_i)$ denotes the neighborhood of node $v_i$. Note that $V_k^p \cup V_k^c = V_k$ and $V_k^p \cap V_k^c = \varnothing$, indicating the union and intersection of the poisoned and clean node sets, respectively. $g_\tau \circ N(v_i)$ represents the poisoned graph structure resulting from embedding trigger $g_\tau$ into the neighborhood of node $v_i$. $\tau$ denotes the target label. $N_\tau = |g_\tau|$ denotes the trigger size and $\triangle_g$ represents the constraint that ensures the trigger size remains within the specified limit. $\rho = \frac{|V_k^p|}{|V_k|}$ represents the poisoning rate, and $\triangle_p$ denotes the budget allocated for poisoned nodes.

In a federated graph backdoor attack, the process of generating triggers and poisoned datasets can be divided into two key steps: trigger generation and trigger injection. The term "trigger" refers to a specific pattern, which has been formally defined as a subgraph in prior work (Zhang et al., 2021b), offering a clear and well-established framework. In our evaluation, we fix poisoning rate $\rho = 0.3$, according to previous work (Wan et al., 2025).

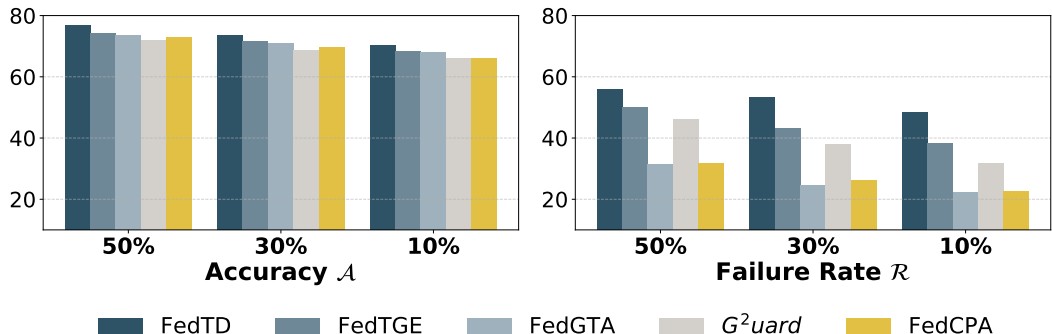

Figure 8: Client selection analysis on the Cora dataset under Non-IID-Louvain setting with a malicious proportion of $\Upsilon = 0.3$ and a trigger type of Renyi.

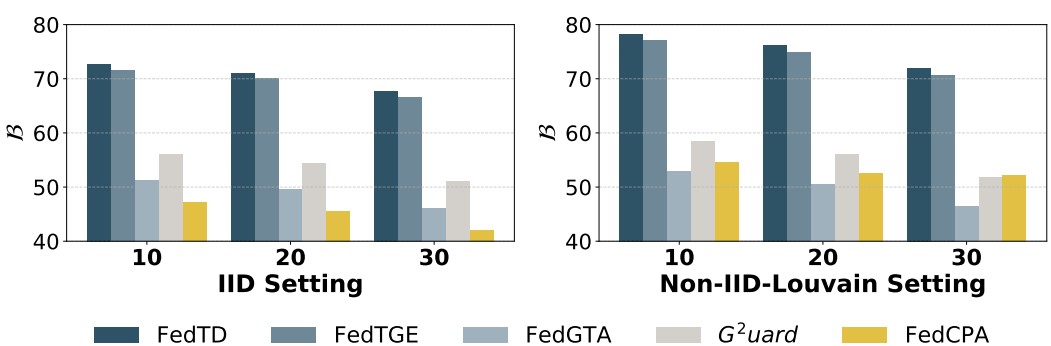

Figure 9: Varying client scales analysis on the Pubmed dataset under both IID and Non-IID-Louvain setting with a malicious proportion of $\Upsilon = 0.3$ and a trigger type of Renyi.

### A.10 CLIENT SELECTION

While many current FGL studies assume full client participation, it is undoubtedly more realistic and efficient to select only a subset of clients. Therefore, we have included experiments based on more practical scenarios using random selection (before training). Specifically, for random selection, we select a subset of clients from all available clients at the beginning of each training round, with participation rates set to 50%, 30%, and 10%. As shown in Figure 7 and Figure 8, FedTD consistently outperforms all baselines under all participation rates. Specifically, we observed a smaller decline in node classification accuracy $\mathcal{A}$ compared to the larger decline in backdoor failure rate $\mathcal{R}$. This is because, in scenarios involving malicious clients, selecting only a subset of clients for participation often results in rounds where few or even no malicious clients are included. Such situations improve accuracy but limit the ability to detect malicious clients effectively.

### A.11 VARYING CLIENT SCALES

Figure 9 illustrates the performance of our FedTD method alongside advanced baselines across varying client scales on the Pumbed dataset. The results indicate that FedSPA consistently achieves superior performance compared to other methods, regardless of changes in client scale. Notably, the number of clients is adjusted proportionally based on the size of the graph.

### A.12 GRAPH OVERLAP STUDY

Real-world FGL often involves entities (nodes) that exist across multiple clients (e.g., a user present in multiple social network databases). Instead of a strict disjoint partition ($V_i \cap V_j = \varnothing$), use an overlapping partition strategy. Allow a specific ratio of nodes (overlap ratio $\delta$) to be shared among $k$ clients. We conduct relevant experiments on the Cora dataset under both IID and Non-IID-Louvain settings with overlap ratios $\delta \in 0.1, 0.2, 0.3$. During local training, clients treat shared nodes as their

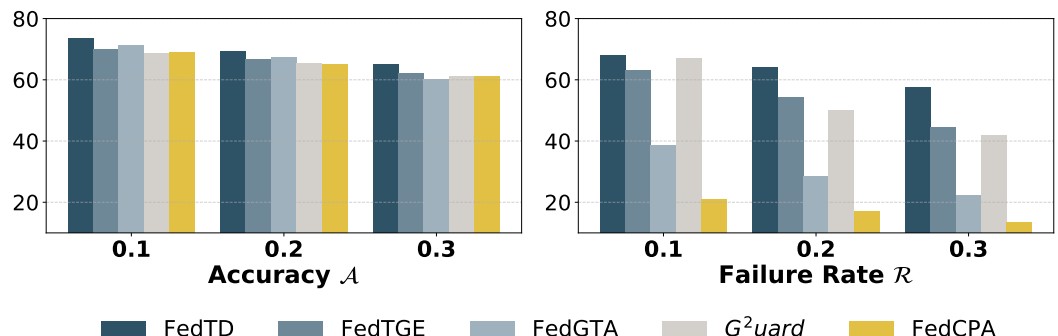

Figure 10: Graph overlap study on the Cora dataset under IID setting with a malicious proportion of $\Upsilon = 0.3$ and a trigger type of Renyi.

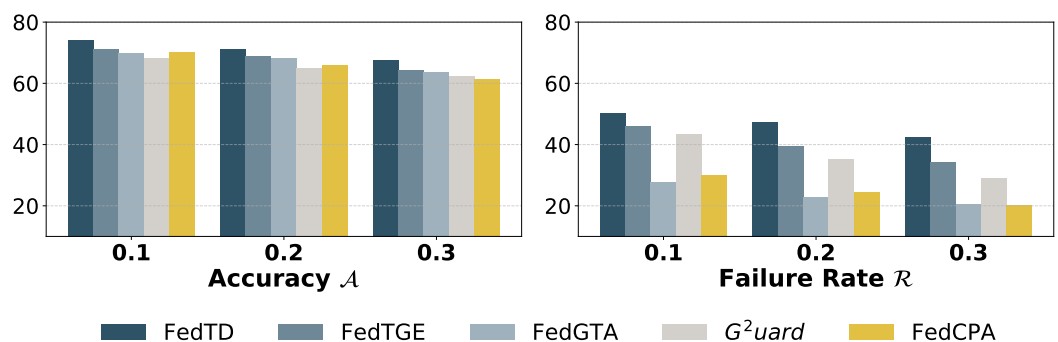

Figure 11: Graph overlap study on the Cora dataset under Non-IID-Louvain setting with a malicious proportion of $\Upsilon = 0.3$ and a trigger type of Renyi.

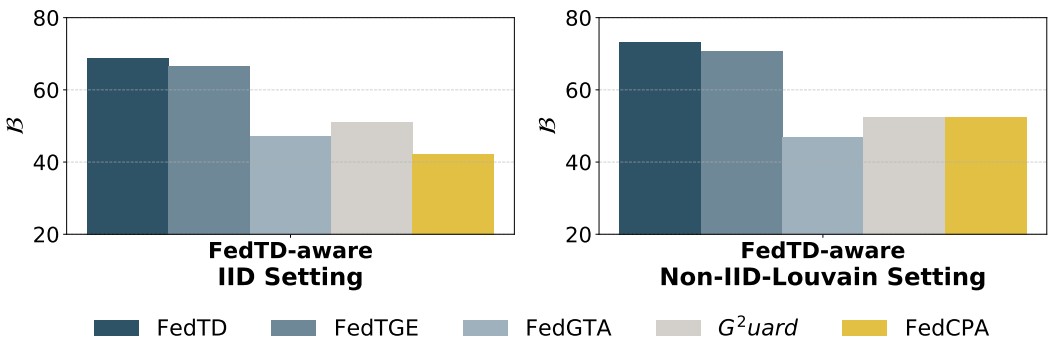

Figure 12: FedTD's defense performance against specific FedTD-aware attack strategy on the Pubmed dataset under both IID and Non-IID-Louvain settings.

own. During aggregation, the server treats the clients as distinct entities (as per the current FedTD design). Figure 10 and Figure 11 show that results for FedTD and all baselines slightly as overlap increases (due to potential label/feature conflicts or redundancy), but FedTD still outperform all baselines.

A.13    DEFENSE AGAINST ADAPTIVE (FEDTD-AWARE) ATTACKS

We further evaluate FedTD's defense performance against specific FedTD-aware attack strategy. We propose a colluding clients forming dense subclusters. The intuition is that if malicious clients are aware that FedTD penalizes outliers in the virtual graph, they may attempt to cluster together to appear more "normal" or "central". Specifically, malicious clients coordinate to minimize the

cosine distance between their estimation vectors ($z_k$) and inject similar trigger patterns into their local training, forcing their embeddings to converge around a target malicious mean. The attackers' goal is to form a dense clique in the server's virtual graph, thereby increasing their degree centrality $d_k$. We conduct relevant experiments on the Pubmed dataset under both IID and Non-IID-Louvain settings. As Figure 12 shows, even if malicious clients align their distributions, their topological features ($T_k$) still differ from benign clients. While the failure rate of FedTD degrade compared to non-adaptive attacks, it remains higher than all baselines.

## A.14 THE USE OF LARGE LANGUAGE MODELS (LLMs)

We utilized large language models (LLMs) in a limited capacity, solely for writing assistance tasks such as grammar correction, style refinement, and table formatting. All suggested changes were carefully reviewed and selectively incorporated by the authors. The scientific content, ideas, analysis, and conclusions presented in this paper are entirely our own. The authors take full responsibility for the paper's content, including any remaining errors or inaccuracies.

