# OpenReview forum: "Topology- and Distribution-aware Backdoor Defense Against Federated Graph Learning"
_ICLR.cc/2026/Conference — Submitted to ICLR 2026_

### Official Review · Reviewer_f9Jx · 2025-10-26

**Soundness:** 3
**Presentation:** 2
**Contribution:** 3
**Rating:** 4
**Confidence:** 3

**Summary:**

The paper presents FedTD , a new robust defense mechanism to counter backdoor attacks in Federated Graph Learning (FGL). Unlike other defenses which the authors describe as 'topology blind’, and as ignoring under and over graph homophily levels across clients, FedTD manages this at the client (TDCE) level by having each client create a fully summarized estimation vector that makes a weighted combination of its data distribution (modeled with an energy-based GCN) and topological features (node degree and centrality). Server side (TDGC) also provides a ‘virtual client graph’ by clustering estimation vectors based on cosine similarity.
The authors’ main contribution on this is the adjustment of similarity score with client homophily levels, which gives a reliable anomalous behavior score. In the virtual graph, malicious clients are identified as low degree outliers, and are given lower weights during model aggregation, nullifying the attack. FedTD under these conditions on five datasets, with small stealthy triggers and varying malicious clients ratios has shown to significantly outperform 14 baselines.  Robustness to hyper-parameters has been shown and the proposed topological features in conjunction with distributional energy and energy shift loss homophily adjustment showed the necessity of each component in ablation studies.

**Strengths:**

- A homophily aware virtual client graph coupled with a topology augmented energy based client estimator (TDCE+TDGC) integrates structure and distribution in a manner not seen in previous defenses within the federated graph learning (FGL) space, like FedTGE.

- Extensive experimentation across five disparate datasets under both IID/Non IID Louvain configurations and varying malicious client ratios demonstrates significant and consistent improvements relative to strong federated learning and federated graph learning baselines, which is further substantiated through ablation studies, convergence training curves, and hyperparameter optimization.

- It uniquely considers backdoor defenses within federated node classification, a significant, under covered threat paradigm to which both topology and homophily are critical, which underpins its applicability to resilient, distributed GNNs.

- The method's lightweight constituents (classical graph topology features, single hidden layer MLP, degree based weighting, N=1 perturbation) imply minimal operational burden and ease of adoption into current federated graph learning workflows.

**Weaknesses:**

-	While the paper claims to be the first integration of topology and distribution for robust FGL, topology aware aggregation and other related FGL works exist (though not for backdoor defense).

-	The paper grids over R and $\tau$ but it’s unclear whether the competing methods receive the same level of tuning or recommended parameters from their works. Furthermore, all experiments utilize a 2 layer GCN for node classification; it’s unclear if GAT/GraphSAGE/heterophily oriented GNNs or other tasks would yield consistency in the gains reported.

-	The formulation explicitly enforces $V=⋃_kV_k$ with $V_i∩V_j=\phi$, ignoring cross-client edges/overlaps common in practical FGL; the paper does not discuss how FedTD would handle inter-client connectivity.

-	The document keeps mentioning a “comprehensive metric B” along with accuracy (A) and backdoor failure rate R without discussing how B is calculated (weighted average, harmonic mean, etc.). This makes it hard to interpret and reproduce.

-	The evaluation considers fixed, randomly placed, size-4 triggers and only changes trigger type and the malicious-client ratio. There is no counter evidence against adversaries who adapt to FedTD signals (e.g., topology-mimicking or homophily-aware triggers), leaving the proof of robustness against stronger attacks unverified.

-	While the method touches on efficiency options, such as using a single-layer MLP for the estimator, it still lacks reported timing and communication costs, as well as scalability studies (clients × graph size), and therefore the deployment cost-benefit remains ambiguous.

**Questions:**

1.	Solve the points hightlighted in weaknesses

2. The experiments appear to use full participation each round with fixed K, yet in practice, FL frequently utilizes partial participation and suffers client churn. What happens with FedTD when some clients only partially attend a few rounds, or when a few clients who are active in some rounds changes continuously? Please include results that demonstrate varying participation rate (e.g., 10/30/50%) and number of clients K and also describe the stability of the virtual graph (edge density/connected components) over rounds in the situations described.

3. Section 3.2 introduces the energy shift loss $L_{ESC}$ with gradients $\nabla_{v_i} S(v_i)$ while Perturb(⋅) includes discrete edge add/remove operations (Eq. 3–4). Please clarify: (i) what variables the gradient is actually taken with respect to (node features only, logits, or a relaxed adjacency); (ii) the full training objective (is cross entropy combined with $L_{ESC}$? with what weight $\lambda$?). A short pseudocode snippet plus a wall time/overhead table would make the method much easier to reproduce and reason about.

---

> ### Author Response · Authors · 2025-11-19
>
> Thanks for your comments. We will address your questions point by point.
>
> > **Weakness 1**
>
> Thank you for raising this point. We fully acknowledge that in the field of Federated Graph Learning (FGL), some existing works have explored topology-aware aggregation methods, which indeed incorporate topological information to improve model performance or handle heterogeneity in graph-structured data.
>
> However, the **key contribution** of our work, **FedTD**, lies in:
>
> 1) **Being the first to systematically integrate topological features with data distribution characteristics specifically for defending against backdoor attacks in FGL**. Existing topology-aware FGL works primarily focus on improving model performance or addressing data heterogeneity, and are not designed to counter backdoor threats.
>
> 2) Our proposed components—**TDCE (Topology- and Distribution-aware Client Estimation)** and **TDGC (Topology- and Distribution-aware Graph Construction)**—leverage not only topological features (e.g., node degree, clustering coefficient) but also energy distributions and homophily levels to achieve finer-grained identification of malicious clients.
>
> Therefore, while topology-aware FGL research does exist, **the integration of topology and distribution for backdoor defense remains a novel contribution of our work**. In the revised version, we have strengthened the discussion of related works in Section 2 to better contextualize FedTD's innovation.
>
> > **Weakness 2**
>
> We added a statement in Section 4.1 (highlighed in blue): We conduct a hyperparameter grid search for all baselines following the ranges suggested in their respective papers.
>
> Thank you for your suggestion. We further examined the impact of replacing GCN with GAT and GraphSage on FedTD's performance. Details can be found in Appendix A.8 in revised version (highlighted in blue).
>
> > **Weakness 3**
>
> Thank you for raising this critical point. Our current work builds upon a common assumption in FGL research where client graph data are completely disjoint. This simplification was adopted to allow us to focus on proposing and validating the core mechanisms of our FedTD against backdoor attacks in a distributed, non-overlapping graph setting.
>
> Although FedTD's formulation does not explicitly handle inter-client edges, its core components **are amenable to incorporating inter-client relational information**.
>
> 1) **In TDCE**: The client's topological feature vector $T_k$ could be enriched with metrics that capture cross-client connectivity (e.g., the degree centrality of nodes that have cross-client links), thereby reflecting both its "internal" topology and "external" associations in the client estimation.
> 2) **In TDGC**: The server-constructed virtual graph is inherently a model of inter-client relationships. If actual inter-client edges exist, they could be leveraged to **initialize or reinforce** the construction of edges in this virtual graph.
>
>
> The primary goal of this paper is to establish, for the first time, the effectiveness of **combining topological and distributional features for defending against backdoor attacks in FGL**. We chose a well-established, foundational setting to demonstrate this core idea clearly and in a controlled manner. Handling overlapping clients and inter-client edges is a significant and complex next step, which we identify as a key direction for future work.
>
> > **Weakness 4**
>
> Thank you for your thoughtful suggestion. We have added an explanation in Section 4.1 (highlighted in blue): $\mathcal{B} = (\mathcal{A} + \mathcal{R})/2$.

---

> ### Author Response · Authors · 2025-11-19
>
> > **Weakness 5**
>
> Thank you for your suggestion. In fact, our experimental setup is already more comprehensive and rigorous compared to most prior related studies. To further address your concerns, we have added experimental results for Opt-GDBA [1] as a trigger, which can be found in Appendix A.7 (highlighted in blue). It is worth noting that for Opt-GDBA, which considers node importance and graph structure when designing triggers, our FedTD demonstrates significantly greater advantages compared to other baselines.
>
> > **Weakness 6**
>
> Thank you for your suggestion. We have provided a complexity analysis as well as an empirical comparison of efficiency. These additions have been included in Section 4.2.5 of the revised version and are highlighted in blue.
>
> > **Question 2**
>
> Thank you for your question. While many current FGL studies assume full client participation, it is undoubtedly more realistic and efficient to select only a subset of clients. Therefore, we have included experiments based on more practical scenarios using random selection (before training). Moreover, dynamically selecting clients before training means that not all clients can participate in the early stages of training. However, since the distinction between malicious and benign clients is based on relative ranking, the absence of certain malicious or benign clients during training does not affect FedTD's ability to identify malicious clients. The consistent performance advantage of FedTD under varying proportions of randomly selected clients demonstrates its robustness and stability. Furthermore, we have also provided experiments on varying client scales. These additions have been included in Appendix A.10 and Appendix A.11 of the revised version and are highlighted in blue.
>
> > **Question 3**
>
> To enable end-to-end gradient flow despite the discrete perturbations, we adopt the following practice:
>
> 1) **Feature Perturbation**: The perturbation applied to node features $x_i$ and their neighbors' features is continuous (adding Gaussian noise), and is thus differentiable with respect to features and model parameters.
> 2) **Structure Perturbation**: The edge addition/removal operations are inherently discrete. In our implementation, we circumvent the need to differentiate through discrete graph structures by randomly sampling perturbation instances. For each node $v_i$, we independently sample $N$ different perturbed graph structure. The computation of $S(v_i)$ and consequently $L_{ESC}$ is performed over this expectation of $N$ samples. The gradients therefore flow through the model's forward passes on these sampled instances, not through the discrete edge operations themselves.
>
>
> In short, the gradients are taken with respect to node features and the randomness from structural perturbations, making the loss function differentiable with respect to the learnable parameters.
>
> We have provided a brief pseudocode in Appendix A.2, and added a runtime/overhead table in our response to your **Weakness 6** concerns.
>
> Refs:
>
> [1] Distributed Backdoor Attacks on Federated Graph Learning and Certified Defenses, CCS 2024.
>
>
>
> We sincerely appreciate your valuable comments and hope our response has adequately addressed your concerns, providing sufficient reasons to raise the score. We are pleased to address any further concerns you may have, and we wish you all the best!
>
> **Friendly reminder: ICLR allows the submission of a revised version during the rebuttal period. Please check revised version directly for more details.**

---

> ### Comment · Reviewer_f9Jx · 2025-11-26
>
> The authors delivered many points successfully. However, I ask the authors to consider the following remaining points:
>
> - **Assumption of Disjoint Graphs:** The authors have clarified that the disjoint-graphs assumption is maintained. While theoretical extensions for inter-client edges were mentioned, experimental results under graph overlap are still missing.
> - **Adaptive Strategies:** While Opt-GDBA results are promising, the paper would benefit from discussing defense performance against specific **FedTD-aware strategies**, such as colluding clients creating dense subclusters.
> - **Loss Formulation:** The combined loss function (specifically the weighting between CE and LESC) needs to be explicitly defined in the main text. The reference to Appendix A.2 is noted, but the explicit formula ensures reproducibility without relying solely on pseudocode interpretation.
>
> Because of the enhancement the authors did and addressing some of the comments, I raised my score to 6; however, it could not be raised more because of the previous comments.

---

> > ### Author Response · Authors · 2025-11-26
> >
> > We sincerely appreciate your constructive feedback and are encouraged by your recognition of our efforts, as reflected in the improved score.
> >
> > We have carefully addressed your remaining three concerns regarding graph overlap, adaptive attacks, and loss formulation. For the first two problems, we will provide the relevant experimental results within the next few days to further enhance the quality of the manuscript. Regarding the last problem, we have added an explicit statement about the combined loss function in the revised version.

---

> ### Author Response · Authors · 2025-11-29
>
> Despite some unexpected circumstances affecting the ICLR rebuttal, we have completed the promised experiments mentioned above to enhance the quality of our work.
>
>
>
> > Graph Overlap Study
>
> Real-world FGL often involves entities (nodes) that exist across multiple clients (e.g., a user present in multiple social network databases). Instead of a strict disjoint partition ($V_i \cap V_j = \varnothing$), use an overlapping partition strategy. Allow a specific ratio of nodes (overlap ratio $\delta$) to be shared among $k$ clients. We conduct relevant experiments on the Cora dataset under both IID and Non-IID-Louvain settings with overlap ratios $\delta \in {0.1, 0.2, 0.3}$. During local training, clients treat shared nodes as their own. During aggregation, the server treats the clients as distinct entities (as per the current FedTD design). Results show that results for FedTD and all baselines slightly degrade as overlap increases (due to potential label/feature conflicts or redundancy), but FedTD still outperforms all baselines.
>
> > Defense Against Adaptive (FedTD-Aware) Attacks
>
> We further evaluate FedTD's defense performance against specific FedTD-aware attack strategy. We propose a colluding clients forming dense subclusters. The intuition is that if malicious clients are aware that FedTD penalizes outliers in the virtual graph, they may attempt to cluster together to appear more "normal" or "central". Specifically, malicious clients coordinate to minimize the cosine distance between their estimation vectors ($z_k$) and inject similar trigger patterns into their local training, forcing their embeddings to converge around a target malicious mean. The attackers' goal is to form a dense clique in the server's virtual graph, thereby increasing their degree centrality $d_k$. We conduct relevant experiments on the Pubmed dataset under both IID and Non-IID-Louvain settings. As results show, even if malicious clients align their distributions, their topological features ($T_k$) still differ from benign clients. While the failure rate of FedTD degrade compared to non-adaptive attacks, it remains higher than all baselines.
>
>
>
> The relevant modifications have been added and highlighted in **Appendix A.12** and **Appendix A.13** of the revised version.
>
> We sincerely appreciate your recognition and further suggestions, and we hope our response has adequately addressed your concerns.

---

### Official Review · Reviewer_PsPP · 2025-10-27

**Soundness:** 3
**Presentation:** 3
**Contribution:** 3
**Rating:** 6
**Confidence:** 3

**Summary:**

The manuscript introduces a topology- and distribution-aware backdoor defense framework FedTD, which integrates topological information and data distribution characteristics to enhance both client- and server-level robustness. At the client level, an energy-based modeling mechanism assigns low energy values to benign clients and high energy values to malicious ones by jointly capturing structural and distributional features. At the server level, a virtual graph is constructed to estimate the maliciousness score of each client while considering homophily levels to ensure the reliability of topological assessment. During aggregation, clients with higher malicious scores are assigned smaller weights, thereby suppressing poisoned updates and improving model resilience. Extensive experiments across diverse FGL scenarios demonstrate that FedTD maintains strong defense capability even under high ratios of compromised clients.

**Strengths:**

This manuscript presents a novel and well-motivated contribution to the filed of FGL. It effectively combines graph topology and data distribution information through two complementary modules TDCE and TDGC. This dual-level design introduces a principled mechanism for distinguishing benign from malicious clients.

Methodologically, this manuscript is technically sound and clearly articulated. It provides formal definitions, derivations, and algorithmic explanations of the energy-based modeling and homophily-aware similarity estimation. The inclusion of homophily level adjustments is particularly insightful, as it captures structural diversity inherent to federated graph settings.

The authors benchmark FedTD against a wide range of baselines, including classical FL defenses and FGL-specific methods, across five datasets under both IID and Non-IID-Louvain settings. Results consistently demonstrate that FedTD outperforms competitors in both accuracy and backdoor resistance, showing robustness across varying malicious client ratios and trigger types. Ablation studies and hyperparameter analyses further confirm the contribution of each component, enhancing the paper’s empirical credibility.

**Weaknesses:**

While the methodology is conceptually strong, the theoretical foundation of the proposed energy-based estimation and the connection between energy distribution and malicious behavior remain largely empirical. The paper does not provide formal guarantees or analytical insights into why the proposed energy–topology fusion should generalize across heterogeneous graph distributions.

The computational and communication overhead introduced by constructing and maintaining the virtual client graph and computing homophily-aware similarities is not analyzed in detail. For large-scale FGL scenarios with numerous clients, the scalability and real-time feasibility of FedTD may be a concern. Additionally, the sensitivity to hyperparameters (e.g., threshold $\tau$, energy perturbation strength, temperature $\gamma$) is only briefly discussed without exploring stability across different environments.

While the experiments cover multiple datasets, they focus solely on node classification. The generality of FedTD to other FGL tasks (e.g., link prediction, graph classification) remains untested. The attack baselines also primarily rely on traditional trigger designs, without evaluating against adaptive or dynamic backdoor strategies, which could better test the method’s robustness.

The paper’s related work discussion, though broad, lacks deeper contrast with recent graph-specific robust learning frameworks (e.g., defense via certified aggregation or representation smoothing). This limits the clarity of FedTD’s position within the broader robustness landscape.

**Questions:**

Please refer to ```weakness``` part.

---

> ### Author Response · Authors · 2025-11-19
>
> Thanks for your comments. We will address your questions point by point.
>
> > **Weakness 1**
>
> The energy-based model used in FedTD is grounded in established principles from energy-based learning. The energy function $E_{\theta(v_i)}$ defined in Eq.2 is derived from the logits of a GCN and approximates the unnormalized log probability of a node belonging to the benign data distribution. Lower energy values indicate higher likelihood under the benign distribution, while higher energies signal deviations-often indicative of malicious perturbations. This aligns with prior work on energy-based out-of-distribution detection, where energy serves as a proxy for data likelihood.
>
> Malicious clients inject triggers that alter both graph topology and node features, leading to anomalous energy distributions. The energy shift loss (Eq. 3) explicitly encourages the model to assign higher energy to perturbed (potentially malicious) samples. This is empirically validated through ablation studies (Section 4.2.2), where removing energy shift loss degrades performance, confirming its role in distinguishing malicious behavior.
>
> While we do not provide formal theoretical guarantees, our extensive experiments across five datasets with varying homophily levels, graph sizes, and distributions (IID and Non-IID) demonstrate consistent performance (Tables 1-2, 4-13). The integration of topological features (e.g., node degree, clustering coefficient) and homophily-aware adjustments (Eq. 7) ensures that FedTD captures structural invariants relevant to backdoor triggers, enhancing generalization. The virtual graph construction in TDGC further leverages graph structural properties (e.g., degree centrality) to identify outliers, which is a principled approach in graph-based anomaly detection.
>
> We agree that formal theoretical analyses—such as convergence guarantees or generalization bounds—would significantly strengthen the work. In the future, we will actively explore this direction.
>
> In summary, while the theoretical foundations are partially implicit in the design, the consistent empirical results across diverse and heterogeneous settings provide strong evidence of the method’s robustness and generalizability.
>
> > **Weakness 2**
>
> Thank you for your suggestion. We have provided a complexity analysis as well as an empirical comparison of efficiency. These additions have been included in Section 4.2.5 of the revised version and are highlighted in blue.
>
> The communication cost is nearly consistent with other works. To address potential concerns from readers, we will include a brief explanation in the revised version. The only additional communication introduced by FedTD is the upload of the client estimation vector $z_k$ from each client to the server. The dimension of $z_k$ is a hyperparameter (we used a small MLP); in our experiments, it was a low-dimensional vector. This constitutes a negligible increase in bandwidth compared to transmitting the entire local model update $w_k^t$.
>
> In Figure 5, we present an analysis of hyperparameter sensitivity. Overall, the performance remains consistently robust across different trigger types and $\Upsilon$ settings. Specifically, for nearly all configurations, setting $\tau = 0.5$ or $\tau = 0.6$ achieves optimal performance. Additionally, to reduce the cost of hyperparameter tuning, we fix $\gamma = 1$ for all scenarios. Based on these observations, FedTD demonstrates low sensitivity to hyperparameters and a relatively low cost of hyperparameter tuning.

---

> ### Author Response · Authors · 2025-11-19
>
> > **Weakness 3**
>
> Thank you for this question. This paper primarily focuses on node-level tasks. However, FedTD is equally applicable to graph classification and link prediction tasks.
>
> We also applied the proposed method to graph classification tasks to further validate its scalability. The results are presented below:
>
> **graph classification**
>
>
> | Methods      | NCI1 (A,R,B)            | PROTEINS_full (A,R,B)   | DD (A,R,B)              |
> | ------------ | ----------------------- | ----------------------- | ----------------------- |
> | FedAvg       | 81.22, 17.59, 49.41     | 73.84, 48.61, 61.23     | 63.84, 16.88, 40.36     |
> | Trimmed Mean | 80.78, 51.59, 66.19     | 74.21, 48.60, 61.41     | 62.04, 11.43, 36.74     |
> | Trim Median  | 79.51, 30.70, 55.11     | 74.77, 48.61, 61.69     | 64.11, 18.33, 41.22     |
> | GA2uard      | 78.70, 42.26, 60.48     | 74.40, 49.21, 61.81     | 64.51, 13.03, 38.77     |
> | Sageflow     | 76.67, 27.48, 52.08     | 74.79, 47.38, 61.09     | 66.11, 14.07, 40.09     |
> | FreqFed      | 76.60, 40.93, 58.77     | 76.05, 52.78, 64.42     | 61.24, 14.46, 37.85     |
> | FedTGE       | 80.13, 64.89, 72.51     | 75.29, 61.23, 68.26     | 65.89, 47.62, 56.76     |
> | FedTD        | 82.08, 64.72, **73.40** | 77.01, 62.15, **69.58** | 67.25, 50.05, **58.65** |
>
> **link prediction**
>
> The attack methods for link prediction are highly similar to those used in node classification, as both involve injecting fake nodes or subgraphs to manipulate the contextual information of local graph data. Consequently, FedTD is theoretically fully applicable to link prediction tasks.
>
> We have added experimental results for Opt-GDBA [1] as a trigger, which can be found in Appendix A.7 (highlighted in blue). It is worth noting that for Opt-GDBA, which considers node importance and graph structure when designing triggers, our FedTD demonstrates significantly greater advantages compared to other baselines.
>
> > **Weakness 4**
>
> Thank you for your suggestion. We have expanded the discussion in the related work section in the revised version (highlighted in blue).
>
> Refs:
>
> [1] Distributed Backdoor Attacks on Federated Graph Learning and Certified Defenses, CCS 2024.
>
>
>
> We sincerely appreciate your valuable comments and hope our response has adequately addressed your concerns, providing sufficient reasons to raise the score. We are pleased to address any further concerns you may have, and we wish you all the best!
>
> **Friendly reminder: ICLR allows the submission of a revised version during the rebuttal period. Please check revised version directly for more details.**

---

> > ### Comment · Reviewer_PsPP · 2025-11-25
> > **Official Comment by Reviewer PsPP**
> >
> > The authors’ response satisfactorily addresses my primary concerns. Nevertheless, leveraging energy-based learning for backdoor defense is not a novel direction[1,2], which limits the incremental contribution of this work. In addition, the complexity analysis is located in Section 4.2.5 rather than Section 4.5.2 as stated. Overall, I will maintain my original score.
> >
> >
> >
> > [1] Wan et al., Energy-based Backdoor Defense Against Federated Graph Learning, ICLR 2025.
> >
> > [2] Gao et al., Energy-based Backdoor Defense without Task-Specific Samples and Model Retraining, ICML 2025.

---

> > > ### Author Response · Authors · 2025-11-25
> > >
> > > Thank you for your reminder and for recognizing our rebuttal. We would like to take this opportunity to emphasize our contributions and innovations. The reference you mentioned, [1] (FedTGE), is one of the key baselines in our evaluation. FedTGE is the first to attempt leveraging energy-based learning for backdoor defense in the field of federated graph learning. However, it lacks consideration of certain inherent characteristics specific to federated graph learning scenarios.
> > >
> > > Our proposed FedTD addresses this by incorporating graph topological features into the energy estimation process and introducing the concept of graph homogeneity (refer to Eq. 7), which is a significant distinction between federated graph settings and traditional federated learning settings.
> > >
> > > As for reference [2] you mentioned, there is a fundamental difference in the setup between it and our proposed FedTD and reference [1] FedTGE.
> > >
> > > Thank you again for your comments and replies, and hope that our further answers can effectively address your concerns. I wish you all the best.

---

### Official Review · Reviewer_WeHY · 2025-10-30

**Soundness:** 2
**Presentation:** 2
**Contribution:** 2
**Rating:** 2
**Confidence:** 4

**Summary:**

This paper proposes FedTD, a defense method against backdoor attacks in federated graph learning. It combines graph topology, data distribution, and homophily information to detect malicious clients and reduce their impact during aggregation. Experiments show that FedTD outperforms existing methods under various datasets and attack scenarios

**Strengths:**

1. The paper clearly points out two major limitations of existing FGL defenses—neglecting topology and homophily differences—with a clear research motivation.

2. The experiments are extensive, covering both IID and Non-IID settings, and comparing with 14 SOTA methods.

**Weaknesses:**

1. The paper lacks a clear threat model. It is recommended that the authors provide a detailed description of this part.

2.  The authors could include models other than GCN for comparison to demonstrate the generality of the proposed defense across different architectures.

3. In the main text, the defense is evaluated against random backdoor attacks rather than SOTA ones, which does not fully demonstrate the effectiveness of the proposed method. It is recommended that the authors test against more advanced backdoor attacks such as Opt-GDBA [Yang et al' 2024].

4. Although efficiency is emphasized, the paper does not provide concrete comparisons of FedTD with baseline methods in terms of computation time or communication cost.

**Questions:**

It is unclear how metric B is calculated; some implementation details should be provided.

---

> ### Author Response · Authors · 2025-11-19
>
> Thanks for your comments. We will address your questions point by point.
>
> > **Weakness 1**
>
> Thank you for your suggestion. We have provided references for threat model and all trigger types in Section 4.1 and included detailed introductions to all trigger types in Appendix A.5. Brief introudction for threat model is provided in Appendix A.9.
>
> > **Weakness 2**
>
> Thank you for your suggestion. We further examined the impact of replacing GCN with GAT and GraphSage on FedTD's performance. Details can be found in Appendix A.8 in revised version (highlighted in blue).
>
> > **Weakness 3**
>
> Thank you for your valuable suggestions. We have added experimental results for Opt-GDBA as a trigger, which can be found in Appendix A.7 (highlighted in blue). It is worth noting that for Opt-GDBA, which considers node importance and graph structure when designing triggers, our FedTD demonstrates significantly greater advantages compared to other baselines.
>
> > **Weakness 4**
>
> Thank you for your suggestion. We have provided a complexity analysis as well as an empirical comparison of efficiency. These additions have been included in Section 4.2.5 of the revised version and are highlighted in blue.
>
> The communication cost is nearly consistent with other works. To address potential concerns from readers, we will include a brief explanation in the revised version. The only additional communication introduced by FedTD is the upload of the client estimation vector $z_k$ from each client to the server. The dimension of $z_k$ is a hyperparameter (we used a small MLP); in our experiments, it was a low-dimensional vector. This constitutes a negligible increase in bandwidth compared to transmitting the entire local model update $w_k^t$.
>
> > **Question 1**
>
> Thank you for your thoughtful suggestion. We have added an explanation in Section 4.1 (highlighted in blue): $\mathcal{B} = (\mathcal{A} + \mathcal{R})/2$.
>
>
>
> We sincerely appreciate your valuable feedback, which has undoubtedly enhanced the rigor and professionalism of our work.  We hope our response has adequately addressed your concerns, providing sufficient reasons to raise the score. We are pleased to address any further concerns you may have, and we wish you all the best!
>
> **Friendly reminder: ICLR allows the submission of a revised version during the rebuttal period. Please check revised version directly for more details.**

---

> ### Author Response · Authors · 2025-11-26
> **Supplement of response to Reviewer WeHY (Part1)**
>
> For your convenience, we have provided the updated experimental results in the comments. You can also find all revised parts in the revised version.
>
>
>
> > For **Weakness 1**
>
> We adopt threat model following previous studies. Assume there are $K$ clients in total, among which $M$ ($M \leq K$) are malicious. Each malicious client independently executes a backdoor attack on its own model. The primary objective of a backdoor attack is to manipulate the model so that it misclassifies specific pre-defined labels (referred to as target labels) within the poisoned data samples, while maintaining high accuracy on clean data. **Attack Knowledge**: In this context, it is assumed that each malicious client has complete knowledge of its own training data and can generate triggers accordingly. This assumption is practical, given that clients have full control over their local data. **Attacker Capability**: A malicious client can inject triggers into its training datasets, subject to predefined constraints such as trigger size and the poisoning rate. The purpose of this is to contaminate the training data. However, the attacker cannot manipulate the server-side aggregation process or interfere with the training processes or models of other clients.
>
> Due to the limited support for formulas on OpenReview, please refer to **Appendix A.9** in the revised version for more detailed formula definitions.
>
>
>
>
> > For **Weakness 2**
>
> We further examined the impact of replacing GCN with GAT and GraphSage on FedTD's performance
>
> | Scenarios        |               | Cora          |               |      |               | Pubmed        |               |      |               | CS            |               |      |               | Physics       |               |      |               | Photo         |               |
> | ---------------- | ------------- | ------------- | ------------- | ---- | ------------- | ------------- | ------------- | ---- | ------------- | ------------- | ------------- | ---- | ------------- | ------------- | ------------- | ---- | ------------- | ------------- | ------------- |
> |                  | $\mathcal{A}$ | $\mathcal{R}$ | $\mathcal{B}$ |      | $\mathcal{A}$ | $\mathcal{R}$ | $\mathcal{B}$ |      | $\mathcal{A}$ | $\mathcal{R}$ | $\mathcal{B}$ |      | $\mathcal{A}$ | $\mathcal{R}$ | $\mathcal{B}$ |      | $\mathcal{A}$ | $\mathcal{R}$ | $\mathcal{B}$ |
> | IID              |               |               |               |      |               |               |               |      |               |               |               |      |               |               |               |      |               |               |               |
> | FedTD(GCN)       | 76.37         | 72.91         | 74.64         |      | 85.09         | 60.31         | 72.70         |      | 85.69         | 71.20         | 78.45         |      | 93.80         | 62.65         | 78.23         |      | 82.50         | 97.04         | 89.77         |
> | FedTD(GAT)       | 77.01         | 70.75         | 73.88         |      | 85.52         | 60.07         | 72.80         |      | 85.90         | 71.31         | 78.61         |      | 93.55         | 62.81         | 78.18         |      | 82.58         | 96.74         | 89.66         |
> | FedTD(GraphSage) | 75.37         | 71.55         | 73.46         |      | 84.62         | 60.00         | 72.31         |      | 85.23         | 70.81         | 78.02         |      | 93.70         | 62.29         | 78.00         |      | 82.56         | 96.89         | 89.73         |
> | Non-IID-Louvain  |               |               |               |      |               |               |               |      |               |               |               |      |               |               |               |      |               |               |               |
> | FedTD(GCN)       | 78.78         | 58.02         | 68.40         |      | 87.21         | 69.07         | 78.14         |      | 89.92         | 73.89         | 81.91         |      | 95.09         | 61.09         | 78.09         |      | 77.91         | 94.69         | 86.30         |
> | FedTD(GAT)       | 78.09         | 58.15         | 68.12         |      | 87.06         | 69.13         | 78.10         |      | 89.12         | 73.34         | 79.18         |      | 95.26         | 61.03         | 78.15         |      | 77.59         | 94.33         | 85.96         |
> | FedTD(GraphSage) | 78.41         | 59.10         | 68.76         |      | 87.30         | 69.41         | 78.22         |      | 88.70         | 73.22         | 80.96         |      | 93.97         | 63.12         | 78.55         |      | 77.84         | 93.95         | 85.90         |
>
> **Table:** Comparison with different GNN backbones with a malicious proportion of $\Upsilon = 0.3$ and a trigger type of Renyi.

---

> ### Author Response · Authors · 2025-11-26
> **Supplement of response to Reviewer WeHY (Part2)**
>
> > For **Weakness 3**
>
> We have added experimental results for Opt-GDBA as a trigger, which can be found in Appendix A.7 (highlighted in blue). It is worth noting that for Opt-GDBA, which considers node importance and graph structure when designing triggers, our FedTD demonstrates significantly greater advantages compared to other baselines.
>
> |Scenarios||Cora||||Pubmed||||CS||||Physics||||Photo||
> |-|-|-|-|-|-|-|-|-|-|-|-|-|-|-|-|-|-|-|-|
> |Metrics|$\mathcal{A}$|$\mathcal{R}$|$\mathcal{B}$||$\mathcal{A}$|$\mathcal{R}$|$\mathcal{B}$||$\mathcal{A}$|$\mathcal{R}$|$\mathcal{B}$||$\mathcal{A}$|$\mathcal{R}$|$\mathcal{B}$||$\mathcal{A}$|$\mathcal{R}$|$\mathcal{B}$|
> |FedAvg|59.68|58.56|59.12||67.94|65.43|66.69||69.17|58.98|64.08||75.12|43.86|59.49||56.34|64.62|60.48|
> |Trimmed Median|58.97|61.18|60.08||67.31|53.34|60.33||70.58|59.64|65.11||72.52|39.91|56.22||64.39|60.12|62.26|
> |Trimmed Mean|65.73|72.91|69.32||66.98|55.31|61.15||69.08|57.19|63.14||74.92|40.99|57.96||63.65|62.93|63.29|
> |FoolsGold|65.75|66.59|66.17||68.23|60.13|64.18||69.34|60.70|65.02||75.67|48.35|62.01||62.15|70.18|66.17|
> |DnC|50.22|68.65|59.44||60.72|54.48|57.60||58.63|75.89|67.26||60.74|59.64|60.19||48.96|77.23|63.10|
> |SageFlow|65.99|72.49|69.24||67.88|67.74|67.81||69.82|58.33|64.08||74.95|50.86|62.91||60.65|56.95|58.80|
> |MMA|63.91|71.33|67.62||67.98|61.72|64.85||67.74|60.17|63.95||75.14|49.08|62.11||62.88|58.16|60.52|
> |RLR|64.05|75.93|*69.99*||68.51|59.67|64.09||69.68|58.79|64.24||75.32|50.19|62.75||62.15|59.16|60.66|
> |Freqfed|66.94|55.76|61.35||68.80|58.10|63.45||70.01|70.32|70.17||74.99|50.56|62.78||63.74|60.61|62.18|
> |FedCPA|64.81|70.44|67.63||67.02|61.72|64.37||69.18|60.75|64.97||75.01|52.40|63.71||63.72|59.73|61.73|
> |G$^2$uard|60.61|74.60|67.61||66.94|60.67|63.80||69.26|60.92|65.09||72.99|51.49|62.24||59.97|58.61|59.29|
> |FedGTA|62.84|64.47|63.66||67.57|61.63|64.60||69.03|60.89|64.96||74.34|48.46|61.40||57.48|62.86|60.17|
> |FGGP|62.45|65.69|64.07||66.76|61.80|64.28||69.51|60.49|65.00||72.59|48.24|60.42||61.83|60.78|61.31|
> |FedTGE|65.31|72.43|68.87||67.10|69.65|*68.38*||70.03|71.91|*70.97*||75.64|66.27|*70.96*||60.92|75.82|*68.37*|
> |FedTD|71.92|80.25|**76.09**||76.52|74.51|**75.52**||78.28|79.03|**78.66**||83.91|73.85|**78.88**||71.55|85.01|**78.28**|
>
> **Table:** Comparison with baselines over three distinct scenarios on five mainstream datasets under IID setting with a malicious proportion of $\Upsilon = 0.3$ and a trigger type of Opt-GDBA.
>
>
>
> |Scenarios||Cora||||Pubmed||||CS||||Physics||||Photo||
> |-|-|-|-|-|-|-|-|-|-|-|-|-|-|-|-|-|-|-|-|
> |Metrics|$\mathcal{A}$|$\mathcal{R}$|$\mathcal{B}$||$\mathcal{A}$|$\mathcal{R}$|$\mathcal{B}$||$\mathcal{A}$|$\mathcal{R}$|$\mathcal{B}$||$\mathcal{A}$|$\mathcal{R}$|$\mathcal{B}$||$\mathcal{A}$|$\mathcal{R}$|$\mathcal{B}$|
> |FedAvg|60.02|58.87|59.45||68.46|65.86|67.16||70.91|60.88|65.90||75.42|48.92|62.17||57.81|62.01|59.91|
> |Trimmed Median|60.94|56.92|58.93||68.73|54.00|61.37||72.19|61.94|67.07||75.13|44.42|59.78||63.78|62.77|63.28|
> |Trimmed Mean|65.86|73.16|69.51||68.64|55.80|62.22||72.70|60.84|66.77||75.64|44.96|60.30||64.97|62.98|63.97|
> |FoolsGold|65.45|67.74|66.60||69.61|61.08|65.34||72.85|61.70|67.28||75.39|52.63|64.01||62.07|71.03|66.55|
> |DnC|49.90|67.77|58.84||61.66|55.72|58.69||60.11|77.73|68.92||60.62|62.51|61.57||50.77|78.98|64.88|
> |SageFlow|65.93|70.07|68.00||69.61|67.87|68.74||71.88|60.74|66.31||75.66|51.98|63.82||61.23|57.75|59.49|
> |MMA|65.79|72.63|69.21||69.60|63.68|66.64||69.36|62.71|66.04||75.54|48.40|61.97||63.34|58.38|60.86|
> |RLR|65.81|77.81|71.81||70.20|59.30|64.75||72.20|60.59|66.40||75.13|50.15|62.64||62.04|59.01|60.53|
> |Freqfed|66.86|56.84|61.85||69.50|59.00|64.25||73.49|71.55|72.52||74.44|53.73|64.09||63.20|62.94|63.07|
> |FedCPA|65.76|71.77|68.77||68.17|62.14|65.16||72.01|62.91|67.46||75.64|53.74|64.69||64.62|60.31|62.47|
> |G$^2$uard|62.87|75.31|69.09||68.62|61.13|64.88||71.95|62.68|67.31||74.31|53.71|64.01||60.93|59.69|60.31|
> |FedGTA|64.85|65.70|65.28||69.23|63.85|66.54||71.95|63.06|67.51||75.19|49.24|62.22||58.88|62.65|60.77|
> |FGGP|64.84|66.35|65.60||67.06|63.11|65.09||71.40|62.85|67.12||74.50|49.05|61.78||62.37|61.90|62.14|
> |FedTGE|65.91|73.16|*69.54*||70.55|73.57|*72.06*||72.95|74.33|*73.64*||75.80|68.18|*71.99*||60.81|77.91|*69.36*|
> |FedTD|73.19|84.61|**78.90**||79.23|81.09|**80.16**||80.28|81.85|**81.07**||82.09|75.66|**78.88**||70.72|84.44|**77.58**|
>
>
> **Table:** Comparison with baselines over three distinct scenarios on five mainstream datasets under Non-IID-Louvain setting with a malicious proportion of $\Upsilon = 0.3$ and a trigger type of Opt-GDBA.

---

> ### Author Response · Authors · 2025-11-26
> **Supplement of response to Reviewer WeHY (Part3)**
>
> > For **Weakness 4**
>
> We analyze the computational complexity of FedTD, which comprises two main components: TDCE and TDGC, examined separately.
>
> For **TDCE**: Total complexity is:
>
> $\mathcal{O}(E \times F \times N) + \mathcal{O}(D \times d^{2}) \approx \mathcal{O}(D \times F \times N).$
>
> where $D$ represent the number of nodes, $F$ the number of features per node, $E$ the number of edges in the local graph, and $d$ is average degree.
>
> For **TDGC**: Total complexity is:
>
> $\mathcal{O}(K^{2} \times L) +  \mathcal{O}(K^{2}) +  \mathcal{O}(K) \approx \mathcal{O}(K^{2} \times L).$
>
> where $K$ denote the number of clients and $L$ the length of energy distributions.
>
> Therefore, overall complexity for FedTD is:
>
> $\mathcal{O}(D \times F) + \mathcal{O}(K^{2}).$
>
> More detailed analysis can be found in **Section 4.2.5** in revised version.
>
>
>
> Furthermore, we conducted empirical experiments to compare the efficiency of FedTD with advanced baselines. The experimental setup remains consistent with previous experiments.
>
> | Dataset | FedCPA  | G$^2$uard | FedGTA  | FGGP    | FedTGE | FedTD  |
> | ------- | ------- | --------- | ------- | ------- | ------ | ------ |
> | Pubmed  | 32.16s  | 38.29s    | 37.21s  | 44.76s  | 26.29s | 28.05s |
> | Physics | 102.27s | 119.22s   | 115.57s | 135.00s | 82.93s | 84.29s |
> | Photo   | 44.19s  | 51.20s    | 49.14s  | 60.09s  | 37.29s | 39.02s |
>
> **Table:** Efficiency analysis for FedTD and baselines.
>
> Notably, the only additional communication introduced by FedTD is the upload of the client estimation vector $z_k$ from each client to the server. The dimension of $z_k$ is a hyperparameter (we used a small MLP); in our experiments, it was a low-dimensional vector. This constitutes a negligible increase in bandwidth compared to transmitting the entire local model update $w_k^t$.
>
>
>
> > This is a supplement to the above rebuttal. All updates can be found in the revised version.
>
> > We sincerely appreciate your valuable comments and hope our response has adequately addressed your concerns, providing sufficient reasons to raise the score. We are pleased to address any further concerns you may have, and we wish you all the best!

---

### Official Review · Reviewer_qwRC · 2025-11-01

**Soundness:** 2
**Presentation:** 3
**Contribution:** 3
**Rating:** 6
**Confidence:** 3

**Summary:**

This paper proposes FedTD, a novel topology- and distribution-aware defense framework against backdoor attacks in Federated Graph Learning (FGL). To effectively identify malicious clients, FedTD introduces a client-side estimation module that jointly leverages data distribution and five complementary topological features to capture structural signatures of backdoor triggers and enhance robustness across diverse attack patterns. At the server level, FedTD constructs a virtual client graph that explicitly accounts for varying homophily levels across local graphs, enabling more reliable similarity measurement and maliciousness scoring. Extensive experiments on multiple real-world graph datasets under both IID and Non-IID settings demonstrate that FedTD significantly outperforms existing baselines, validating the effectiveness of its integrated design.

**Strengths:**

1. The propsoed method integrates five complementary topological features like node degree, PageRank, clustering coefficient into client estimation to capture structural signatures of backdoor triggers and enhancing robustness against diverse attack patterns.
2. The method  proposes the topology and distribution aware graph construction to address the limitation of existing approaches, which fail to account for the varying homophily levels across clients.
3. The effectiveness of the proposed module is validated on both real-world and synthetic graph datasets.

**Weaknesses:**

1.	The threshold τ is crucial for constructing the virtual graph and the paper lacks analysis of the choice of τ  in methods and experiments.
2.	The methodology section introduces several design choices like constructing virtual graphs and selecting five specific topological features, however, the paper lacks a complexity analysis of the proposed method.
3.	The experiment lacks a comparative analysis of running efficiency between the proposed method and the baselines.
4.	Although the paper provides an anonymous code link, the corresponding files are missing.

**Questions:**

1. The structural information aware is crucial to the proposed method. However, the paper lacks detailed explanation and justification regarding why  "Node Degree, Local Clustering Coefficient, Degree Centrality, PageRank Score, and Standard Deviation of Neighbor Degrees" can help "mitigate the potential noise." or further clarify why these particular features were chosen as structural-aware descriptors?
2. Could the proposed method be compared with more recent baseline methods?

---

> ### Author Response · Authors · 2025-11-19
>
> Thanks for your comments. We will address your questions point by point.
>
> > **Weakness 1**
>
> Thank you for your question. You may have missed the hyperparameter analysis of $\tau$ provided in our paper. Please refer to Figure 5, with the corresponding grid search range detailed in Section 4.1.
>
> > **Weakness 2 & Question 1**
>
> Thank you for the insightful comment. We selected these five topological features—Node Degree, Local Clustering Coefficient, Degree Centrality, PageRank, and Std of Neighbor Degrees—because they collectively capture complementary structural aspects crucial for identifying backdoor triggers, which often manifest as localized subgraphs with anomalous connectivity. (refer to Appendix A.1). For example:
>
> - **Node Degree & Degree Centrality** detect nodes with unusually high/low connectivity (potential trigger hubs).
> - **Clustering Coefficient** spots irregular local structures (e.g., non-native cliques or overly sparse patterns).
> - **PageRank** identifies nodes with anomalously high influence due to injected edges.
> - **Std of Neighbor Degrees** highlights 'bridge' nodes connecting disparate graph regions—a common trigger role.
>
> Their effectiveness is validated in our ablation study (refer to Section 4.2.2), where removing topological features (w/o-T) consistently degraded performance. The complexity analysis and empirical efficiency comparison have been included in Section 4.2.5 of the revised version and are highlighted in blue.
>
> > **Weakness 3**
>
> Thank you for your suggestion. We have provided a complexity analysis as well as an empirical comparison of efficiency. These additions have been included in Section 4.2.5 of the revised version and are highlighted in blue.
>
> > **Weakness 4**
>
> Thank you for your suggestion. We have identified this as a bug on the anonymous repository platform. The code is now accessible.
>
> > **Question 2**
>
> Our evaluation encompass a range of models, from traditional FL defenses to the latest FGL-specific defenses, which sufficiently demonstrate the superiority of FedTD. For instance, FedTGE is one of the most recent works in this area that has been accepted by top-tier venues. If you have any suggestions for additional comparison baselines (preferably open-source), we are glad to incorporate them into our experiments as soon as possible.
>
>
>
> We sincerely appreciate your valuable comments and hope our response has adequately addressed your concerns, providing sufficient reasons to raise the score. We are pleased to address any further concerns you may have, and we wish you all the best!
>
> **Friendly reminder: ICLR allows the submission of a revised version during the rebuttal period. Please check revised version directly for more details.**

---

> ### Author Response · Authors · 2025-11-26
> **Supplement of response to Reviewer qwRC**
>
> For your convenience, we have provided the updated experimental results in the comments. You can also find all revised parts in the revised version.
>
> > Complexity Analysis
>
> We analyze the computational complexity of FedTD, which comprises two main components: TDCE and TDGC, examined separately.
>
> For **TDCE**: Total complexity is:
>
> $\mathcal{O}(E \times F \times N) + \mathcal{O}(D \times d^{2}) \approx \mathcal{O}(D \times F \times N).$
>
> where $D$ represent the number of nodes, $F$ the number of features per node, $E$ the number of edges in the local graph, and $d$ is average degree.
>
> For **TDGC**: Total complexity is:
>
> $\mathcal{O}(K^{2} \times L) +  \mathcal{O}(K^{2}) +  \mathcal{O}(K) \approx \mathcal{O}(K^{2} \times L).$
>
> where $K$ denotes the number of clients and $L$ the length of energy distributions.
>
> Therefore, overall complexity for FedTD is:
>
> $\mathcal{O}(D \times F) + \mathcal{O}(K^{2}).$
>
> More detailed analysis can be found in **Section 4.2.5** in the revised version.
>
>
>
> Furthermore, we conducted empirical experiments to compare the efficiency of FedTD with advanced baselines. The experimental setup remains consistent with previous experiments.
>
> | Dataset | FedCPA  | G$^2$uard | FedGTA  | FGGP    | FedTGE | FedTD  |
> | ------- | ------- | --------- | ------- | ------- | ------ | ------ |
> | Pubmed  | 32.16s  | 38.29s    | 37.21s  | 44.76s  | 26.29s | 28.05s |
> | Physics | 102.27s | 119.22s   | 115.57s | 135.00s | 82.93s | 84.29s |
> | Photo   | 44.19s  | 51.20s    | 49.14s  | 60.09s  | 37.29s | 39.02s |
>
> **Table:** Efficiency analysis for FedTD and baselines.
>
> Notably, the only additional communication introduced by FedTD is the upload of the client estimation vector $z_k$ from each client to the server. The dimension of $z_k$ is a hyperparameter (we used a small MLP); in our experiments, it was a low-dimensional vector. This constitutes a negligible increase in bandwidth compared to transmitting the entire local model update $w_k^t$.
>
>
>
> > This is a supplement to the above rebuttal. All updates can be found in the revised version.
>
> > We sincerely appreciate your valuable comments and hope our response has adequately addressed your concerns, providing sufficient reasons to raise the score. We are pleased to address any further concerns you may have, and we wish you all the best!

---

### Author Response · Authors · 2025-11-29
**Summary of the rebuttal and revisions**

Dear AC, SAC, and PC,

We sincerely thank all reviewers for their insightful and constructive comments, despite some unexpected circumstances that arose during the rebuttal period.

We fully recognize the significant responsibilities and workload that the AC has had to manage under these unexpected circumstances. To facilitate the process, we have prepared a concise summary of our rebuttal, the reviewers' comments, and the revisions we have made during this period.

Prior to the unexpected circumstances, two reviewers (PsPP and f9Jx) expressed support for our rebuttal, indicating that we had addressed nearly all of their concerns. Reviewer f9Jx raised our score and posed further suggestions, to which we have now provided a comprehensive response. Although further discussion with the reviewers was not possible due to the circumstances, we believe our updated responses adequately address these remaining points.

For the other two reviewers who were unable to engage further, we note that most of their concerns overlap with those already resolved in the discussion with PsPP and f9Jx.

In summary, we have incorporated the following key additions and revisions:

- Introduction to graph learning in related work (Section 2.2)
- Complexity analysis (Section 4.2.5)
- Threat model introduction (Appendix A.9)
- Analysis of different graph backbones (Appendix A.8)
- A new trigger method—Opt-GDBA (Appendix A.5)
- Study on client selection (Appendix A.10)
- Analysis of varying client scales (Appendix A.11)
- Graph overlap analysis (Appendix A.12)
- Defense Against Adaptive (FedTD-Aware) Attacks (Appendix A.12)
- Additionally, numerous minor revisions and clarifications have been made throughout the manuscript, all highlighted in the revised version.

To ensure that the AC, SAC, PC, and all reviewers can easily verify the revisions, all changes have been thoroughly documented in the revised version and are clearly marked in blue.

Thank you once again for your time and consideration.

Warm regards,

Authors of ICLR 5565

---

### Meta-Review · Area_Chair_k4uR · 2026-01-07

**Summary:**

1. Contribution: Although the paper provides a novel integration of topology and distribution for federated graph learning defenses, the core technique of energy-based backdoor defense is not new (Reviewer PsPP explicitly notes prior works such as Wan et al., ICLR 2025, Gao et al., ICML 2025), limiting the overall incremental contribution of the method.

2. The paper primarily relies on empirical comparisons, lacking in-depth theoretical analysis or insights explaining why the proposed method achieves better performance (as specifically highlighted by Reviewer PsPP). This reduces the overall significance and interpretability of the results.

3. Incomplete Threat Model and Initial Evaluation Setup:
The original submission lacked a clear and detailed threat model (pointed out by Reviewer WeHY) and did not initially evaluate against advanced or adaptive backdoor attacks, such as Opt-GDBA (Reviewers WeHY and f9Jx). Although these concerns have been partially addressed during rebuttal, the initial omission weakens confidence in the robustness claims.

4. Limited Generalizability and Experimental Scope:
Multiple reviewers (qwRC, WeHY, f9Jx) raised concerns regarding the narrowness of experimental evaluations—initially focused primarily on node classification tasks using a simple two-layer GCN backbone. Although additional results with GAT and GraphSAGE were provided, broader experimental validations (e.g., more complex tasks, other attack scenarios, realistic overlapping graphs) were only partially addressed during rebuttal.

5. Complexity and Practicality Issues:
Reviewers qwRC, PsPP, and f9Jx raised concerns regarding the method’s computational and communication overhead, especially considering realistic, large-scale federated graph learning scenarios. Although the authors later provided a complexity analysis and some efficiency comparisons, practical considerations for larger scales, varying client participation rates, and real-world deployments were only superficially covered and remain not fully convincing.

**Reviewer Concerns:**

Concerns addressed by rebuttal:
* Expanded empirical evaluation (additional GNN backbones, Opt-GDBA attack, graph-overlap scenarios).
* Clearly defined threat model and clarified experimental metric definitions.
* Provided detailed complexity and efficiency analyses, addressing scalability concerns.

Concerns remaining:
* Incremental novelty remains a significant concern, given closely related prior work (Reviewer PsPP).
* Real-world generalization (adaptive attackers, dynamic client participation, practical scalability) is still not convincingly demonstrated (Reviewers f9Jx, WeHY).
* The work largely relies on empirical performance comparisons, with limited deeper analysis or theoretical insight into why the proposed method works, which weakens the overall contribution (Reviewer PsPP).

**Reviewer Scores:**

Reviewer qwRC (score: 6): Would likely maintain their positive assessment, as several minor experimental concerns were addressed, though deeper theoretical justification remains unresolved.

Reviewer WeHY (score: 2): Likely to maintain their negative score, as significant concerns about the threat model clarity, advanced attack evaluation, and representativeness were not fully resolved.

Reviewer PsPP (score: 6): Would likely maintain their score, given that experimental improvements were satisfactory, yet fundamental concerns regarding theoretical novelty and contribution remain unaddressed.

Reviewer f9Jx (score: 4): Would likely slightly improve their score given the added experiments and clarifications, but critical issues about adaptive attacks, deeper analysis, and generalization remain outstanding.

Overall, while the rebuttal resolved many experimental details, major concerns about theoretical depth, threat model clarity, and robustness against adaptive attacks remain outstanding.

---

### Decision · Program_Chairs · 2026-01-26

Reject